# An Interactive Framework for Finding the Preferred Trade-off in Differential Privacy

## Abstract

Differential privacy (DP) is the gold standard for privacy-preserving analysis but introduces a fundamental trade-off between privacy guarantees and model performance. Selecting the optimal balance is a critical challenge, framed as a multi-objective optimization (MOO) problem of discovering the Pareto front and eliciting a decision-maker's preference. While interactive MOO offers a solution, standard approaches—which model objectives separately and rely on simple pairwise feedback—are suboptimal for DP because they do not utilize the problem structure. In this work, we propose a method, **PACE** (**P**rivacy-**A**ccuracy **C**urve **E**licitation), that exploits two key properties to reduce this inefficiency. First, we leverage the fact that the theoretical privacy guarantee naturally serves as a constraint: maximizing accuracy for a specific privacy level generates a solution on the Pareto front. Second, to efficiently model this curve, we theoretically derive the trade-off shape for regularized logistic regression, revealing a characteristic S-curve. This motivates us to model the Pareto front using a sigmoidal function. We empirically demonstrate its effectiveness across studied DP settings. This model allows us to replace less efficient pairwise comparisons with a richer interaction scheme where decision-makers directly select their most preferred solution from the hypothetical trade-off curve. Experiments on differentially private logistic regression and deep transfer learning across six datasets show that PACE converges to the most preferred trade-off with fewer model evaluations and interactions than baselines.

## 1 Introduction

Differential privacy (DP) (Dwork et al., 2006) is a rigorous privacy-preserving framework that has become the de-facto standard for protecting sensitive data, particularly in the context of training deep learning models. Its application introduces a fundamental trade-off between the strength of the privacy guarantee (e.g., privacy budget $\varepsilon$ in $(\varepsilon, \delta)$-DP (Dwork et al., 2006), $\rho$ in $\rho$-zCDP (Bun & Steinke, 2016; Dwork & Rothblum, 2016) and $\mu$ in $\mu$-GDP (Dong et al., 2022)), and the model's performance (e.g., accuracy in classification tasks). Navigating this trade-off presents a dual challenge for practitioners. First, the decision-maker must choose an appropriate value for the privacy level itself, a critical and context-dependent decision (Dankar & El Emam, 2012; Dwork et al., 2019). Second, for a fixed privacy level, one often needs to perform hyperparameter optimization (HPO) over various training parameters to achieve the best accuracy. While significant research has focused on the second challenge (Koskela & Kulkarni, 2023; Panda et al., 2024; Liu & Bu, 2025), the first key question—how to choose the privacy level in a principled manner—still remains a hurdle. Simply relying on conventional values (e.g., $\varepsilon = 1$) is arbitrary and fails to account for the unique context of each application. An overly conservative privacy level may render a model useless for its intended purpose (e.g., failing to detect a medical condition), while an overly permissive one can lead to privacy breaches, eroding user trust and incurring regulatory penalties (Dwork et al., 2019; Nanayakkara et al., 2022). Avent et al. (2020) formulate this trade-off as a multi-objective optimization (MOO) problem to generate the full Pareto front. But this exhaustive approach may become computationally prohibitive for large-scale models. Therefore, a systematic method is needed to help decision-makers navigate this trade-off and select a configuration that balances privacy preferences/needs with performance requirements for deploying the differentially private models.

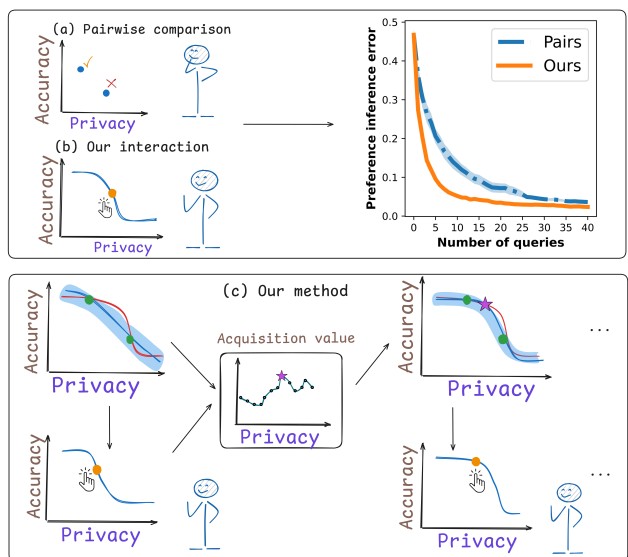

Figure 1: An overview of our interactive framework for finding the most preferred privacy-accuracy trade-off. (Top Panel) Unlike standard pairwise comparisons (a), our interaction (b) elicits richer feedback by asking the user to select their ideal solution on a hypothetical curve. This leads to faster convergence in learning user preferences. (Bottom Panel) Our method alternates between updating the preference learning model based on their choice (orange point) and updating the model (blue line) of the true Pareto front (red line) based on an acquisition function to select the next privacy level to evaluate (purple star).

A more practical approach to solve this is interactive multi-objective optimization (Branke, 2008; Gunantara, 2018), which concurrently explores both the Pareto front—the set of *Pareto optimal* trade-offs—as well as the decision-maker's preferences. This allows it to converge on the single optimal solution that aligns with their needs, without the unnecessary cost of recovering the entire Pareto front. While standard interactive methods have been studied in non-DP settings (Astudillo & Frazier, 2020; Lin et al., 2022; Ozaki et al., 2024), they function as generic, black-box approaches, making them less efficient when applied to DP, as they fail to exploit the unique underlying structure and suffers from two major drawbacks. First, they build surrogate models that map the hyperparameter space to the objectives (privacy and accuracy in our case) to locate the Pareto front. Second, they typically rely on simple pairwise comparisons ("is A better than B?") (Chu & Ghahramani, 2005), which often provide limited feedback per query and, hence, are sample inefficient.

In contrast, our approach leverages two key properties of DP—the theoretical privacy guarantee naturally serves as a constraint: the constrained optimization problem at a fixed privacy level yields a Pareto-front solution; in practice we approximate this solution via HPO. Moreover, we empirically observe that across commonly used DP training pipelines studied in this work, the Pareto front can be well approximated by a sigmoidal curve, transitioning from a noise-dominated performance floor to a non-private accuracy ceiling.

We exploit these findings by proposing a more efficient interactive framework specifically tailored to navigate the privacy-accuracy trade-off in DP (Figure 1). Concretely, we transform multi-objective optimization into multiple single objective optimization with constraints and directly model the Pareto front as a function mapping the privacy level to the best accuracy. Each outcome of this function is determined by a HPO process constrained to a fixed privacy level, ensuring it approximates a Pareto-optimal solution. We ground this modeling choice in both theory and practice: we derive the theoretical trade-off shape for logistic regression and demonstrate empirically that DP models we study follow a sigmoidal trajectory. This insight reveals that the front exhibits a characteristic S-shape, enabling us to employ the Bayesian inference technique with a sigmoidal prior to efficiently approximate the curve from sparse observations. Finally, we replace standard pairwise comparisons in preference learning with a more informative interaction scheme: we present the hypothetical trade-off curves drawn from the posterior and ask decision-makers to select their preferred solution on each curve. This provides richer feedback per query than the simple binary preference learned from a pairwise comparison, reducing the number of interactions needed to identify the preferred trade-off.

Altogether we make the following contributions:

1. We derive a theoretical characterization of the privacy-accuracy trade-off for logistic regression, which motivates modeling the Pareto front with S-shaped functions (Section 4.1).

2. We propose a Bayesian framework that explicitly models the Pareto front, enabling curve-based rather than pointwise preference elicitation (Section 4.2.)

3. We provide a new interactive method: **PACE** (**P**rivacy-**A**ccuracy **C**urve **E**licitation) that guides decision-makers to the privacy level aligned with their preferences, transforming this choice into a structured optimization process, and improves the efficiency of both approximating the Pareto front and learning decision-makers' preferences compared to existing baselines (Section 4.4).

4. We demonstrate that our method outperforms the compared baselines in sample efficiency, requiring fewer model evaluations and interactions to converge on the preferred trade-off across multiple private machine learning tasks (Section 5).

## 2 Background

We begin this section with differential privacy in Section 2.1, which establishes the fundamental privacy-accuracy trade-off we aim to solve. We then introduce multi-objective optimization in Section 2.2, the mathematical lens through which we formalize the trade-off in differential privacy. Finally, we cover interactive multi-objective optimization in Section 2.3, which forms the basis of our contribution

### 2.1 Differential Privacy

Differential privacy (DP) is the gold standard for privacy-preserving analysis, but introduces a fundamental trade-off between privacy guarantee (e.g., $\varepsilon$ in $(\varepsilon, \delta)$-DP (Dwork et al., 2006), $\rho$ in $\rho$-zCDP (Bun & Steinke, 2016; Dwork & Rothblum, 2016) and $\mu$ in $\mu$-GDP (Dong et al., 2022)) and the resulting model's performance (e.g., accuracy in classification tasks). We take $(\varepsilon, \delta)$-DP for illustration.

**Definition 2.1** (($\varepsilon, \delta$)-Differential Privacy)**.** A randomized mechanism $\mathcal{A}$ is $(\varepsilon, \delta)$-differentially private (Dwork et al., 2006) if for any two neighboring datasets $D \in \mathcal{D}$ and $D' \in \mathcal{D}$ differing in only one record (adding or removing), and for any $S \subseteq \mathcal{S}$ in the output space $\mathcal{S}$,

$$P[\mathcal{A}(D) \in S] \le e^{\varepsilon} \times P[\mathcal{A}(D') \in S] + \delta. \tag{1}$$

The parameter $\delta$ is an additive slack term in the bound allowing for a slight relaxation of the strict indistinguishability and usually is set to be a sufficiently small number. When $\delta = 0$, $(\varepsilon, \delta)$-DP becomes pure $\varepsilon$-DP. The parameter $\varepsilon$ is called the privacy budget. It controls the strength of privacy guarantee provided in Definition 2.1: a smaller $\varepsilon$ provides more protection by ensuring that the probability distributions of the mechanism's output on any two neighboring datasets are statistically closer. This increased indistinguishability makes it more difficult for an adversary to infer the presence of any single individual in the dataset, thus enhancing privacy but degrading accuracy, creating an inevitable trade-off.

Though we take $(\varepsilon, \delta)$-DP for illustration, this trade-off between privacy and accuracy also exists for other definitions of DP. Therefore, in this work, we generalize the privacy guarantee using one parameter, $p$, which we refer to as the **privacy level**. To frame the trade-off as a consistent maximization problem, we define our privacy level, $p$, such that **a higher value corresponds to a stronger privacy guarantee.** This allows our framework to treat both privacy and accuracy as objectives to be maximized. For common DP definitions, this can be achieved through a simple transformation. For instance, in the case of $(\varepsilon, \delta)$-DP, where a smaller $\varepsilon$ means stronger privacy, we define the privacy level $p = -\log(\varepsilon)$ in this work.

Differentially private stochastic gradient descent (DP-SGD) (Rajkumar & Agarwal, 2012; Song et al., 2013; Abadi et al., 2016) and differentially private Adam (DP-Adam) are two widely used privacy-preserving optimization algorithms, derived from the standard SGD (Robbins & Monro, 1951) and Adam (Kinga & Ba, 2015) optimizers. To enforce privacy, these private optimizers first clip the per-sample gradients, which limits the maximum influence of any single data point and bounds the sensitivity. Afterwards, they add carefully calibrated statistical noise to these clipped gradients, obscuring individual contributions. While this

clip-then-noise procedure enables private training, it introduces a complex trade-off between the privacy level and model accuracy. The noise required for privacy, which depends on the hyperparameters, typically reduces model accuracy, and the choice of training hyperparameters (e.g., learning rate, clipping threshold, batch size, number of epochs) for a fixed privacy level further determines the final accuracy. Formally, we define our two objectives: privacy $f_p : \boldsymbol{\Theta} \to \mathbb{P}$, which maps a set of hyperparameter to its privacy level, and accuracy $f_\alpha : \boldsymbol{\Theta} \to \mathbb{A}$, which maps a set of hyperparameter to the resulting model accuracy (Avent et al., 2020).

To achieve the best possible model performance for a given dataset $D$, a DP algorithm $\mathcal{A}$, and a privacy level $p \in \mathbb{P}$, practitioners must carefully tune the training hyperparameters, $\boldsymbol{\theta}$, from the search space $\boldsymbol{\Theta}$. This process, known as hyperparameter optimization (HPO), can be formalized as a constrained optimization problem in DP. The goal is to find the optimal set of training hyperparameters that satisfies the privacy constraint and maximizes the model's accuracy:

$$h(p) = \max_{\boldsymbol{\theta}} f_\alpha(\boldsymbol{\theta}; \mathcal{A}, D) \ s.t. \ f_p(\boldsymbol{\theta}; \mathcal{A}, D) \geq p. \tag{2}$$

Thus, $h(p)$ is a mapping from the privacy level $p$ to the corresponding best accuracy obtained by HPO process. If the privacy level $p$ changes for the given task, the training hyperparameters also need be re-optimized to maximize the model's accuracy $f_\alpha$ under the new privacy constraints.

A significant body of work has focused on the HPO process in DP. For instance, one can transfer hyperparameters tuned on a smaller amount of data (Koskela & Kulkarni, 2023; Sander et al., 2023), extrapolate from hyperparameters tuned at smaller $\varepsilon$ (Panda et al., 2024), perform fully differentially private hyperparameter tuning (Liu & Talwar, 2019; Papernot & Steinke, 2022), or even train using methods that eliminate the need for hyperparameter tuning altogether (Liu & Bu, 2025).

## 2.2 Multi-Objective Optimization

The trade-off between privacy and accuracy in DP can be framed as a multi-objective optimization (MOO) problem. Denoting $s$ objectives with $\boldsymbol{f} = (f_1, \ldots, f_s)$ where $f_i : \boldsymbol{X} \to Y_i$, MOO is formalized as

$$\arg \max_{\boldsymbol{x} \in \boldsymbol{X}} \{f_1(\boldsymbol{x}), f_2(\boldsymbol{x}), \ldots, f_s(\boldsymbol{x})\}. \tag{3}$$

As noted, in DP the hyperparameters are the inputs $\boldsymbol{X} := \boldsymbol{\Theta}$. Privacy $f_p : \boldsymbol{\Theta} \to \mathbb{P}$ and accuracy $f_\alpha : \boldsymbol{\Theta} \to \mathbb{A}$ are the two objectives. Typically, there is no single solution that maximizes all objectives. Instead, there is a *Pareto front* of all non-dominated solutions (Branke, 2008).

**Pareto Optimality** For a pair $(\boldsymbol{x}, \boldsymbol{x}')$, we say "$\boldsymbol{x}$ weakly dominates $\boldsymbol{x}'$" if $\boldsymbol{x}$ is no worse than $\boldsymbol{x}'$ in all objectives, $f_i(\boldsymbol{x}) \geq f_i(\boldsymbol{x}')$ for all $i \in \{1, \ldots, s\}$. If at least one of the inequalities is strict, we say "$\boldsymbol{x}$ dominates $\boldsymbol{x}'$". If $\boldsymbol{x}$ is not (weakly) dominated by any other $\boldsymbol{x}'$ in the domain, $\boldsymbol{x}$ is called (weakly) Pareto-optimal. *(Weak) Pareto front* is a set of (weakly) Pareto optimal solutiont.

**Multi-objective Bayesian Optimization (MOBO)** efficiently locates the Pareto front when objective functions are expensive, black-box evaluations. MOBO treats MOO as a sequential design strategy utilizing two components: a probabilistic surrogate model and an acquisition function.

**Bayesian inference** is used to maintain a probabilistic distribution over the unknown objectives. Specifically, we assume a prior (in this case $P(\boldsymbol{f})$) – typically a Gaussian Process (GP, Williams & Rasmussen (1995)) – and apply the Bayes-rule to infer the posterior given some data $\{(\boldsymbol{x}_n, \boldsymbol{y}_n)\}_{n=1}^{N}$ and build the surrogate:

$$P(\boldsymbol{f} \mid \{(\boldsymbol{x}_n, \boldsymbol{y}_n)\}_{n=1}^{N}) \propto P(\boldsymbol{f}) \prod_{i,n}^{N} P(y_{i,n} \mid f_i, \boldsymbol{x}_n). \tag{4}$$

Based on the GP posterior, an **acquisition function** $\alpha(\mathbf{x}|\{(\boldsymbol{x}_n, \boldsymbol{y}_n)\}_{n=1}^{N})$ is maximized to select the next query to efficiently identify the Pareto front. Common acquisition functions include expected hypervolume

improvement (Yang et al., 2019; Daulton et al., 2020) or information-theoretic approaches (Belakaria et al., 2019; Suzuki et al., 2020). The objective $\boldsymbol{f}$ is then evaluated at the new query, the dataset is updated, and the posterior is recomputed iteratively. For example, Avent et al. (2020) employs hypervolume-based probability of improvement as the acquisition function to learn the trade-off between $\varepsilon$ and classification error in DP.

$c$-**Constraint Method** [1] A well-known strategy to finding the Pareto front is to select one of the objective functions to be optimized, and convert the others into constraints (Branke, 2008):

$$\max \ f_l(\boldsymbol{x}) \ s.t. \ f_i(\boldsymbol{x}) \geq c_i, \ i = 1, \ldots, s, \ i \neq l. \tag{5}$$

The solution of Equation (5) can be proven always to be weakly Pareto optimal. In addition, a unique solution can be proven to be Pareto optimal (Branke, 2008; Mavrotas, 2009; Pirouz & Khorram, 2016). For problems with many objectives, defining feasible lower bounds for these constraints in Equation (5) can be challenging. However, in our setting with only two objectives (privacy and accuracy), this method is powerful and straightforward. We select accuracy as the objective to maximize and treat privacy as the constraint, which aligns naturally with the definition of differential privacy. In practice, the hyperparameter search space $\Theta$ is closed and bounded, which guarantees that a maximum accuracy exists for any feasible privacy level.

## 2.3 Interactive MOO: Preference Learning

Because Pareto optimal solutions have no objective ranking, ultimately a decision-maker needs to pick the preferred one. In fact, it is inefficient to first explore the whole Pareto front to only pick one at the end. Instead, *interactive MOO* aims to learn the decision-makers' preferences *while* simultaneously exploring the Pareto front, in order to find the best solution as efficiently as possible. This task has two components: (i) learning the objective functions $\boldsymbol{f}$ and (ii) learning the preferences over the objectives $\boldsymbol{y} = (y_1, \ldots, y_s)$, where $y_i = f_i(\boldsymbol{x})$. We already discussed the first in Section 2.2, and will consider the second below.

Let $U(\boldsymbol{y}; \boldsymbol{w})$ be a utility function that quantifies a decision-marker's preference over objectives $\boldsymbol{y}$, parameterized by the preference weights $\boldsymbol{w}$. To find the trade-off that maximizes this utility[2], a scalarization function is used to convert the vector of multiple objectives into a single scalar value. Chebyshev scalarization function (Miettinen & Mäkelä, 2002) is widely used in MOO as the utility function (Ozaki et al., 2024; Ungredda & Branke, 2023), as it can identify any trade-off on the Pareto front by varying the weights:

$$U(\boldsymbol{y}; \boldsymbol{w}) := \min\left(\frac{y_1}{w_1}, \ldots, \frac{y_s}{w_s}\right), \text{with} \sum_{i=1}^{s} w_i = 1. \tag{6}$$

A fundamental premise of preference learning is that decision-makers are unable to provide the parameters of this function, or report the utility values $U(\boldsymbol{y}; \boldsymbol{w})$ of a query $\boldsymbol{x}$ explicitly (Chu & Ghahramani, 2005; Fürnkranz & Hüllermeier, 2010). Instead, the most common approach is to ask the decision-maker to pick their preferred solution among $q$ options $\{\boldsymbol{y}_j\}_{j=1}^{q}$ (Astudillo et al., 2023; De Peuter et al., 2024). Then, given a user model of how people make such decisions, we can learn the underlying utility function $U$ from the decision-maker's choices (González et al., 2017; Astudillo et al., 2023).

The most widely used model—and the one adopted in our work—is the Boltzmann-rational model (Luce, 1959; Jeon et al., 2020; Yamagata et al., 2024). This model says the probability of choosing an item $\boldsymbol{y}$ among a set of candidates $\{\boldsymbol{y}_j\}_{j=1}^{q}$, given the preferences weight $\boldsymbol{w}$, is proportional to its value:

$$P(\boldsymbol{y} \mid \{\boldsymbol{y}_j\}_{j=1}^{q}, \boldsymbol{w}) = \frac{\exp(U(\boldsymbol{y}; \boldsymbol{w})/T)}{\sum_{j}^{q} \exp(U(\boldsymbol{y}_j; \boldsymbol{w})/T)}, \tag{7}$$

where the temperature coefficient $T$ controls the rationality. A lower $T$ means higher rationality, causing a more deterministic choice, while a higher $T$ means lower rationality, causing a more random and noisy choice.

---

[1]The terminology "$\varepsilon$-constraint method" is usually used in MOO, but here it would become confused with the privacy budget $\varepsilon$. Hence we call the approach here "$c$-constraint method".

[2]Whereas utility is traditionally measured only by model performance (e.g., accuracy) in DP literature, we define utility jointly over privacy and accuracy.

Equation (7) is a statistical model for the likelihood from a Bayesian perspective. In particular, given a dataset $\{\boldsymbol{y}^*, \{\boldsymbol{y}_j\}_{j=1}^q\}$ of the selected solution $\boldsymbol{y}^*$ among candidates $\{\boldsymbol{y}_j\}_{j=1}^q$, the posterior over preferences follows from the Bayes-rule given a prior $P(\boldsymbol{w})$:

$$P(\boldsymbol{w} \mid \{\boldsymbol{y}^*, \{\boldsymbol{y}_j\}_{j=1}^q\}) \propto P(\boldsymbol{w})P(\boldsymbol{y}^* \mid \{\boldsymbol{y}_j\}_{j=1}^q, \boldsymbol{w}). \tag{8}$$

**Active Learning of Preferences** Preference learning is also a sequential design strategy utilizing two components: a probabilistic surrogate model and an acquisition function. After modeling the posterior (Equation 8), the next thing is to decide which pairs of trade-offs to query the decision-makers with, for which we use Knowledge Gradient (KG) (Frazier et al., 2009)—a one-step Bayes-optimal acquisition function. It selects the pair $(\boldsymbol{y}_a, \boldsymbol{y}_b)$ that maximizes the expected improvement in utility as the next query when $q = 2$:

$$\max \mathrm{KG}_t(\boldsymbol{y}_a, \boldsymbol{y}_b) := \max_{\boldsymbol{\beta}} \mathbb{E}[U_{t+1}^* - U_t^* \mid \text{query} = (\boldsymbol{y}_a, \boldsymbol{y}_b)], \tag{9}$$

where $U_t^*$ is the maximum expected utility before the query, and the expectation is taken over the possible outcomes $\boldsymbol{\beta}$ of the user's choice for that pair $(\boldsymbol{y}_a, \boldsymbol{y}_b)$. This iterative process continues to query the decision-maker with the new pair and update the posterior of $\boldsymbol{w}$ until the utility function is sufficiently refined to identify the preferred trade-off on the Pareto front.

## 3 Problem Statement

We operate under the "*trusted curator*" threat model that is common for practical deployments of DP (Avent et al., 2020) and is a standard setting in Dwork & Roth (2014), where a trusted internal team has legitimate access to the sensitive data and is responsible for the entire model development and selection process. The adversary is an external party who only observes the single, final model that is ultimately deployed. Under this threat model, the privacy guarantee protects the individuals in the training dataset against inference from the final released model, not against the trusted internal curator. In this paper, the terms "*user*" and "*decision-maker*" refer specifically to practitioners (e.g., model developers), not the individuals contributing sensitive data. Our goal is to guide these practitioners in selecting a preferred privacy level $p$ for deploying the DP models.

With these assumptions, we formulate the task of balancing privacy $p \in \mathbb{P}$ and accuracy $\alpha \in \mathbb{A}$ in DP as a MOO problem. Formally, we aim to identify hyperparameters $\boldsymbol{\theta} \in \boldsymbol{\Theta}$ that induce an accuracy value $f_\alpha : \boldsymbol{\Theta} \to \mathbb{A}$ and privacy level $f_p : \boldsymbol{\Theta} \to \mathbb{P}$ that best align with the decision-maker's preference. We assume the decision-maker has a latent utility function $U : (\mathbb{P}, \mathbb{A}) \to \mathbb{R}$ parameterized by $\boldsymbol{w}$, which quantifies the overall satisfaction with a given privacy-accuracy trade-off $(p, \alpha)$. Then the goal turns to find the trade-off on the Pareto front that maximizes this utility function. Therefore, our central research problem is to design a sample-efficient framework that solves the following optimization problem to find the preferred trade-off:

$$\max_{\boldsymbol{\theta}} \ U(f_p(\boldsymbol{\theta}), f_\alpha(\boldsymbol{\theta}); \boldsymbol{w}) \tag{10}$$

with the objectives $(f_p, f_\alpha)$ and preference weights $\boldsymbol{w}$ unknown.

## 4 Methodology

To solve Equation (10), we propose an interactive MOO framework: **PACE** (**P**rivacy-**A**ccuracy **C**urve **E**licitation), specifically tailored to the structure of DP. Unlike standard interactive methods that treat objectives as black boxes—building generic surrogate models and relying on information-poor pairwise comparisons—PACE exploits two properties of the DP trade-off to improve efficiency in the settings we study.

First, we leverage the fact that the theoretical differential privacy guarantee in DP naturally serves as a direct optimization constraint. By fixing a privacy level $p$ and maximizing accuracy via HPO, we can formulate a constrained optimization problem whose exact solution $(p, h(p))$ lies on the Pareto front. This allows us to model the Pareto front as a continuous function $h : \mathbb{P} \to \mathbb{A}$ mapping privacy to the best achievable accuracy.

Second, we exploit the insight that in the tasks we study, $h(p)$ exhibits a characteristic S-shape. Rather than modeling complex individual objective functions to locate the Pareto front, we **directly model the Pareto front** itself. Motivated by the theoretical derivation from regularized logistic regression, we use a sigmoidal function as a surrogate to approximate the trade-off curve. This surrogate model enables a novel interaction scheme, allowing decision-makers to select their preferred trade-off directly from hypothetical curves.

In Section 4.1, we introduce our motivation of sigmoidal priors and our Bayesian model of the Pareto front and how we use the privacy level constraint optimization to gather the data necessary for inference. In Section 4.2, we discuss how to model the decision-maker feedback when querying on hypothetical Pareto fronts. Section 4.3 describes how we are able to solve both the tasks of preference and Pareto front learning in a single decision, based on a knowledge-gradient acquisition function. Lastly, the whole framework is summarized in Section 4.4.

## 4.1 Pareto Front Modeling

The driving insight of this work is the observation that the Pareto front $h : \mathbb{P} \to \mathbb{A}$ is well-approximated by analytical S-shaped functions in the settings we study, enabling the use of parametric surrogates to model the front directly. Here we provide theoretical support and empirical evidence for this insight.

**Trade-off between privacy and accuracy of logistic regression**   We start with pure $\varepsilon$-DP and use a well-understood mechanism—output perturbation for regularized logistic regression (Chaudhuri & Monteleoni, 2008)—as a concrete case study to generate and model this trade-off curve between $\varepsilon$ and accuracy. Following the Algorithm 1 in Chaudhuri & Monteleoni (2008), let $(x_1, y_1), \ldots, (x_n, y_n)$ be a set of labeled points over $\mathbb{R}$ with $x_i$ uniformly sampled from $[-1, 1]$. We compute the coefficient $\xi$ obtained by the logistic regression with a regularization constant $\lambda$. Adding noise $\eta$ to $\xi$ gives the noisy coefficient $\xi_{noisy} = \xi + \eta$ with $\eta \sim \mathrm{Lap}(S_f/\varepsilon)$. $S_f$ is the sensitivity of function $f$, which is at most $\frac{2}{n\lambda}$ for logistic regression (Chaudhuri & Monteleoni, 2008).

Here we derive our theoretical formulation of $h(\varepsilon)$, starting with the probability of a correct prediction.

The derivation for a positive class is

$$
\begin{aligned}
P((\xi + \eta)x \geq 0) &= P(\eta x \geq -\xi x) \\
&= P(Z \geq -\xi x) \\
&= 1 - F_Z(-\xi x) \\
&= 1 - 0.5 \cdot \exp\left(-\frac{|\xi|}{S_f}\varepsilon\right). \quad (11)
\end{aligned}
$$

The derivation for a negative class is

$$
\begin{aligned}
P((\xi + \eta)x \leq 0) &= P(\eta x \leq -\xi x) \\
&= P(Z \leq -\xi x) \\
&= F_Z(-\xi x) \\
&= 1 - 0.5 \cdot \exp\left(-\frac{|\xi|}{S_f}\varepsilon\right). \quad (12)
\end{aligned}
$$

The last equation is due to $Z = \eta x \sim \mathrm{Lap}((S_f/\varepsilon)|x|)$. Thus the theoretical accuracy for any single point $x$ is $h(\varepsilon, x) = 1 - 0.5 \cdot \exp(-C\varepsilon)$, where $C = \frac{|\xi|}{S_f}$. The expectation accuracy over the entire data distribution is

$$
\begin{aligned}
\mathbb{E}[h(\varepsilon)] &= \int_{-1}^{1} h(\varepsilon, x) \cdot P(x) dx \quad (13) \\
&= h(\varepsilon) \cdot \int_{-1}^{1} P(x) dx \\
&= 1 - 0.5 \cdot \exp(-C\varepsilon) \\
&= 1 - 0.5 \cdot \exp(-C\exp(\tilde{\varepsilon})),
\end{aligned}
$$

where $\tilde{\varepsilon} = \log(\varepsilon)$. This is a special sigmoidal curve—Gompertz function (Winsor, 1932), which is asymmetric: the curve approaches its right-side asymptote much more gradually than its left-side asymptote. The general form of the Gompertz function with parameters is

$$
h(\varepsilon; L, k, b, c) = \frac{L}{\exp(k \exp(-c \log \varepsilon))} + b. \quad (14)
$$

By taking $p = -\log \varepsilon$, we can obtain

$$
h^g(p; L, k, b, c) = \frac{L}{\exp(k \exp(cp))} + b. \quad (15)
$$

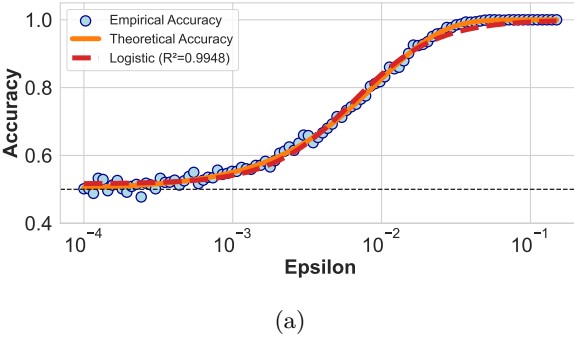 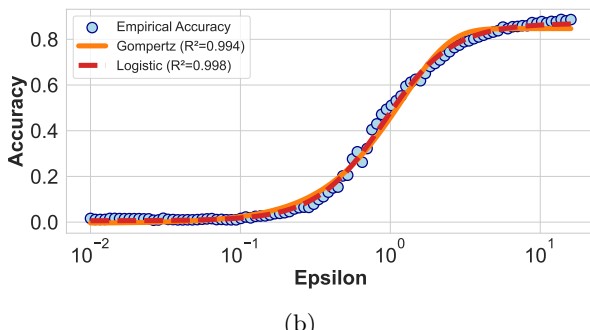

(a)  (b)

Figure 2: (a) The theoretical (Gompertz) and empirically fitted (Logistic) privacy-accuracy trade-off for differentially private logistic regression. The Logistic curve presents a good fit to theoretical accuracy. (b) The Gompertz and Logistic functions fitting to the privacy-accuracy trade-off data for differentially private transfer learning on CIFAR100 with $\delta = 10^{-5}$. Both curves effectively capture the trend of the observed trade-off data.

Next we collect empirical data on the model's performance across a range of privacy budgets $\varepsilon \in [0.0001, 0.15]$ to approximate the Pareto front. Our theoretical derivation for differentially private logistic regression shows under the stated assumptions that the resulting trade-off follows a Gompertz function. The Gompertz curve is a member of the broader family of S-shaped functions, all of which are characterized by a slow initial growth, a rapid increase, and a final saturation towards a plateau.

Based on this intuition, we also model the observed S-shaped relationship with a more general and widely-used functional form—the Logistic function with parameters $\boldsymbol{\beta} = (L, b, k, c)$ to model the trade-off curve:

$$h(\varepsilon; L, k, b, c) = \frac{L}{1 + \exp(-k(\log \varepsilon - c))} + b. \tag{16}$$

By taking $p = -\log(\varepsilon)$, we can obtain

$$h^l(p; L, k, b, c) = \frac{L}{1 + \exp(k(p - c))} + b. \tag{17}$$

Figure 2a shows the Logistic fitted results and indicates that the Logistic function provides a good approximation of the Pareto front in logistic regression. We then generalize this observation to a broader class of differentially private models.

**Pareto front modeling in DP models**   While the formal derivation of this S-shaped relationship is for logistic regression under $\varepsilon$-DP, we view it as evidence of a broader structural pattern that may extend to other differentially private learning. This interpretation is motivated not by model-specific mechanisms, but by general features of the privacy-accuracy trade-off. Specifically, the trade-off curve is bounded by two asymptotes: a performance floor at high privacy levels (e.g., $\varepsilon \to 0$), where injected noise dominates the signal, and a performance ceiling at low privacy levels (e.g., $\varepsilon \to \infty$), where the model approaches its non-private accuracy. Between these regimes, improvements in accuracy typically exhibit diminishing returns as suggested by our empirical results across the studied pipelines: initial relaxations of the privacy can yield substantial accuracy gains as the signal becomes recoverable, while subsequent gains taper off as performance saturates. Consistent with this structural behavior, sigmoidal functions provide an effective surrogate for modeling the Pareto front in the settings we study.

In Figure 2b, we present the trade-off data after performing HPO for each privacy level of differentially private transfer learning on CIFAR100. The details of HPO and the model can be found in Section 5.3. We use both Gompertz and Logistic functions to fit the trade-off data, which gives comparably good fitting results. Our intention was not to claim that the theoretical derivation directly applies to DP-SGD. Rather, the results suggest that this trade-off is well-approximated by S-shaped functions empirically.

Based on our theoretical and empirical findings, we propose to model the Pareto front $h_{\boldsymbol{\beta}} : \mathbb{P} \to \mathbb{A}$, parameterized by $\boldsymbol{\beta}$, between privacy and accuracy in the differentially private models we study using

S-shaped functions as surrogates. In practice, the accuracy of any single model is a random variable due to stochastic elements in the training process and the noise inherent to the differential privacy mechanism.

We adopt a probabilistic approach to account for this variability. In particular, we assume noisy observations are drawn from a Gaussian distribution whose mean is the expected accuracy predicted by the S-curve. This gives us the following likelihood: $\alpha \sim \mathcal{N}(h(p;\boldsymbol{\beta}), \sigma^2)$, where $\sigma^2$ is the variance of the observation noise.

Then, assuming a prior over the parameters ($P(\boldsymbol{\beta})$, $P(\boldsymbol{\sigma})$) and given observed data $\{p_n, \alpha_n\}_{n=1}^N$ with $\alpha_n$ being the maximum accuracies after HPO for $p_n$, standard Bayes-rule gives us the posterior:

$$P(\boldsymbol{\beta}, \sigma \mid \{p_n, \alpha_n\}_{n=1}^N) \propto P(\boldsymbol{\beta})P(\sigma)\prod_n^N \mathcal{N}(\alpha_n \mid h(p_n); \boldsymbol{\beta}, \sigma^2). \tag{18}$$

We employ two S-shaped functions to model the Pareto front $h$ between the privacy $p$ and the accuracy $\alpha$: PACE-G with a Gompertz function (Equation 15) and PACE-L with a Logistic function (Equation 17). Gompertz function comes from the theoretical derivation of the trade-off curve for regularized logistic regression. Logistic function is introduced as a more general and widely used surrogate. The point is to demonstrate that the framework is not tied to a specific S-shaped parameterization.

Each solution on the empirical trade-off curve is obtained from an HPO process conducted for a fixed privacy level $p$, and is therefore intended to approximate a Pareto-optimal solution under the constrained formulation.

## 4.2 Interactive Preference Elicitation

Having established a method to construct a surrogate model of the privacy-accuracy Pareto front, $h_{\boldsymbol{\beta}}$, we now address the subsequent challenge: eliciting a decision-maker's preferences to identify their preferred trade-off. As opposed to querying pairwise comparisons—the most common approach—we propose to exploit the fact that we have a model of the Pareto front to gather richer feedback. In particular, we present *hypothetical* Pareto fronts to the decision-maker, parameterized by $\boldsymbol{\beta}$, and ask them to pick their favorite trade-off $\boldsymbol{y}^* = (p, \alpha)$ on it to elicit the informative feedback about their preferences.

This proposal naturally leads to the central modeling question: **how do we formally interpret the decision-maker's choice?** We adopt the Boltzmann rational user model (Equation 7) and assume that the user has some latent utility function $U(\boldsymbol{y}; \boldsymbol{w})$ to quantify the quality of a trade-off, and that the likelihood of picking a trade-off is proportional to this utility value. To extend this model to handle continuous choices, we argue that *the decision-maker interprets the presented front as a (technically infinite) set of trade-offs*. Hence, we discretize the continuous front $h_{\boldsymbol{\beta}}$ into a set of $q$ points $\{\boldsymbol{y}_j\}_{j=1}^q$.

With this, we have a user (likelihood) model to interpret the decision-maker's trade-off choices. Now, learning the decision-maker's preference is the task of Bayes inference given a prior $P(\boldsymbol{w})$ and preference data $\{\boldsymbol{y}_m^*, \boldsymbol{\beta}_m\}_{m=1}^M$ (Equation 8).

## 4.3 Acquisition Functions

So far, we have discussed Bayesian inference over the Pareto front (Section 4.1) and decision-maker preference elicitation (Section 4.2). This section introduces how to pick which 1) hypothetical Pareto fronts with parameters $\boldsymbol{\beta}$ to show to the decision-maker and 2) constraints $p$ to do (constrained) optimization on to find candidate trade-offs $\boldsymbol{y} = (p, \alpha)$ on the estimated Pareto front that can help maximize the utility. While interactive MOO methods typically solve this problem using two different acquisition functions, one for each posterior, we instead follow Ungredda & Branke (2023) and use a single knowledge-gradient acquisition function across both steps, optimizing it with respect to utility.

Knowledge-gradient (KG; Frazier et al., 2009) selects the next evaluation by maximizing the expected utility increase in the value of the optimal solution, given the information gained from a single new observation. When we have observed a set of $M$ preference observations $\{\boldsymbol{y}_m^*, \boldsymbol{\beta}_m\}_{m=1}^M$ and $N$ Pareto optimal observations

$\{\boldsymbol{y}_n = (p_n, \alpha_n)\}_{n=1}^N$, the expected utility of some trade-off $\boldsymbol{y}$ is computed by

$$\mathbb{E}_{\boldsymbol{\beta}, \boldsymbol{w}}[U(\boldsymbol{y})] = \int_{\boldsymbol{\beta}, \boldsymbol{w}} U(\boldsymbol{y}; \boldsymbol{w}) P(\boldsymbol{w} \mid \{\boldsymbol{\beta}_m, \boldsymbol{y}_m^*\}_{m=1}^M) P(\boldsymbol{\beta} \mid \{\boldsymbol{y}_n\}_{n=1}^N), \tag{19}$$

where the posteriors $P(\boldsymbol{w} \mid \ldots)$ is given by Equation (8), and $P(\boldsymbol{\beta} \mid \ldots)$ denotes the marginal posterior over $\beta$, with $\sigma$ integrated out from Equation (18). The largest expected utility is then

$$U_{M,N}^* := \max_p \mathbb{E}_{\boldsymbol{\beta}, \boldsymbol{w}}[U(p, h_{\boldsymbol{\beta}}(p); \boldsymbol{w})] \tag{20}$$

based on the current posteriors.

Therefore, we maximize the KG with respect to the posterior over the preferences to find the most informative curve to present to the decision-maker:

$$\boldsymbol{\beta}_{M+1} \leftarrow \arg\max_{\boldsymbol{\beta}} \mathbb{E}[U_{M+1,N}^* - U_{M,N}^* \mid \boldsymbol{\beta}_{M+1} = \boldsymbol{\beta}]. \tag{21}$$

Similarly, when picking the most informative constraint $p_{N+1}$ to optimize with, we maximize the KG with respect to the posterior over the Pareto front:

$$p_{N+1} \leftarrow \arg\max_p \mathbb{E}[U_{M,N+1}^* - U_{M,N}^* \mid p_{N+1} = p]. \tag{22}$$

### 4.4 Algorithm

The proposed algorithm is summarized in Algorithm 1. The final output is the estimated preferred trade-off that best aligns with the decision-maker's preferences. The computation of acquisition functions (Equations 21 and 22) can be found in Appendix F. For the experiments presented in this paper, we adopt a direct interleaving scheme, and sequentially perform one model evaluation to refine our estimate of the Pareto front, followed by one interaction to update our model of the decision-maker's preferences. This maintains a balanced allocation on both learning objectives throughout the process. Our acquisition function framework also supports adaptive interleaving (Ungredda & Branke, 2023): after evaluating the acquisition functions from both procedures, the algorithm compares their values to determine which action to take and updates the corresponding posterior distribution accordingly.

---

**Algorithm 1** Interactive algorithm to identify the most preferred trade-off.

---

**Input:** Training dataset $\mathcal{D}$. DP-model $\mathcal{M}$. Search range of hyperparameters $\boldsymbol{\Theta}$. Priors of $\boldsymbol{w}$ and $\boldsymbol{\beta}$: $P(\boldsymbol{w})$ and $P(\boldsymbol{\beta})$, number of iterations NUM.
**for** $t = 1, \ldots, \text{NUM}$ **do**
  **if** Perform Preference Elicitation: **then**
    Select the next curve to present $\boldsymbol{\beta}_{M+1} \leftarrow$ Equation (21).
    Observe decision-maker's choice $\boldsymbol{y}_{M+1}^*$ on the front $\boldsymbol{\beta}_{M+1}$.
    Update posterior over preference $\boldsymbol{w} \leftarrow$ Equation (8).
  **end if**
  **if** Perform Pareto Front Modeling: **then**
    Select the next privacy level $p_{N+1} \leftarrow$ Equation (22).
    HPO for $p \geq p_{N+1}$ to get $\alpha_{N+1} \leftarrow$ Equation (2).
    Update posterior on Pareto front $\boldsymbol{\beta} \leftarrow$ Equation (18).
  **end if**
**end for**
**Output:** The most preferred trade-off $(p^*, \alpha^*)$.

---

## 5 Experiments

Section 5.1 lists the experiments configurations. Section 5.2 presents the experiments on finding the preferred trade-off of differentially private logistic regression. Section 5.3 contains the experiments on finding the

preferred trade-off of differentially private deep transfer learning problems. Then, in Appendix B, we present an ablation study on preference learning and another one on Pareto front modeling, to investigate sample efficiency in both tasks. Lastly, we compare our utility function with other linear utility functions to justify our choice in Appendix B.3. All results report the average and the standard error of 30 runs.

### 5.1 Experiments Configurations

**Hyperparameter Optimization**   For the inner-loop HPO task of finding the maximal accuracy for a given privacy level $p$ (Equation 2), we require an efficient method for optimizing an expensive black-box function. To this end, we select Bayesian Optimization (BO) due to its well-documented sample efficiency (Garnett, 2023). The search range of hyperparameters can be found in Table 1. We employ a standard, non-private version of BO. Under our *trusted curator* threat model, the intermediate results of our interactive optimization process—including all HPO evaluations and the generated Pareto front—are considered confidential to the internal team. Therefore, we do not account for the cumulative privacy cost of the iterative search, as these exploratory steps are never exposed to the adversary. Recent work (Xiang et al., 2024; Pradhan et al., 2025) has investigated the impact of HPO on privacy and found no statistically significant evidence that it increases a modern deep learning model's vulnerability to membership inference attacks.

However, it is crucial to note that our interactive framework can in principle be combined with other HPO methods. One could easily substitute other methods, such as a more efficient algorithm (Liu & Bu, 2025), without altering the core logic of the proposed method. This modularity allows our framework to be readily adapted to different computational constraints and development environments.

Table 1: Search bounds for hyperparameters during Bayesian optimization.

| Parameter | LogReg+SGD | Fine-Tuning+Adam |
|---|---|---|
| **Batch size** | [8, 512] | $[192, |\mathcal{D}|]$ |
| **Learning rate** | [5e-4, 5e-2] | [10e-5, 0.1] (log) |
| **Clipping threshold** | [0.1, 4] | [10e-4, 100] (log) |
| **Epochs** | [1, 64] | 40 |

**Experimental Setup**   Here we list the experiments setup and parameters. We split 10% of the training set $\mathcal{D}$ to form a validation set and use the model's performance on this set as the optimization objective for hyperparameter tuning. We run 20 iterations of BO using the Optuna library (Akiba et al., 2019) with the BoTorch sampler (Balandat et al., 2020) for each fixed theoretical differential privacy guarantee. Afterward, we conduct a final training run on the full $\mathcal{D}$ with the optimal hyperparameters found during tuning and evaluate the final accuracy on the test set for each fixed theoretical differential privacy guarantee. For a fair comparison, we allow each baseline to evaluate 20 hyperparameter configurations at every evaluation step.

We model the decision-maker's choices with Chebyshev utility functions and a Boltzmann-rational model with parameter $T = 0.2$ (recall Sections 2.3 and 4.2). PACE exhibits robustness to the choice of $T$. We present the sensitivity analysis in Appendix C.1. The true (simulated) weights are sampled from a Dirichlet distribution parameterized by $(2, 2)$ (same as Ozaki et al., 2024), which is also used as the prior for the solution methods and shows that PACE can infer the preferred trade-off across a diverse range of preferences. We assume the decision-maker discretizes the trade-off with $q = 100$ points. We show the sensitivity analysis of $q$ in Appendix C.2 to demonstrate that PACE is robust to the choice of $q$. Table 2 in Appendix A lists the prior distributions for parameters of Logistic and Gompertz functions that we use in all experiments.

We use the Opacus (Yousefpour et al., 2021) library to achieve DP guarantee under $(\varepsilon, \delta)$-DP and fix $\delta = 10^{-5}$. For accounting, we employ the PRV accountant (Gopi et al., 2021) that ships with Opacus. We normalize both the accuracy $\alpha$ and privacy level, $p = -\log(\varepsilon)$, to a common $[0, 1]$ scale using a min-max transformation. This process ensures that both objectives are comparable for preference learning, with 1 representing the most desirable outcome for each objective. This normalization is performed on a user pre-defined range of the privacy budget, $[\varepsilon_{\min}, \varepsilon_{\max}]$. The discussion of the rationale and practical guidance for selecting this range are detailed in Section 7. Since this normalization is invertible, any solution in the scaled space can be mapped directly back to its original $\varepsilon$ value.

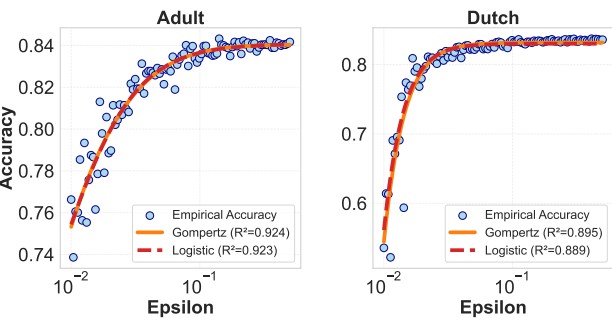

Figure 3: Logistic and Gompertz functions fitted to the empirical privacy-accuracy trade-off for DP logistic regression on the Adult and Dutch datasets with $\delta = 10^{-5}$. Both curves effectively capture the trend of the observed trade-off data.

**Evaluation Metrics** In total 20 steps, we adopt a direct interleaving scheme and sequentially perform one model evaluation at step $t$ and one interaction at step $t + 1$. We report *error* in preference inference and *regret* at step $t$. The error in preference estimation is the expected mean-squared error under the posterior over the preference weights $\boldsymbol{w}$ (Ozaki et al., 2024):

$$\boldsymbol{w}^t_{\text{error}} = \mathop{\mathbb{E}}_{\boldsymbol{w}^t} \|\boldsymbol{w}_{\text{true}} - \boldsymbol{w}^t\|_2. \tag{23}$$

*Regret* is defined as the difference between the utility of the true optimal trade-off, $\boldsymbol{y}^*_{\text{true}}$, and the (true) utility of the privacy level $p^t$ that maximizes the utility under the current posteriors: $p^t := \arg\max_p \mathbb{E}_{\boldsymbol{w}, \boldsymbol{\beta}}[U((p, h_\beta(p)); \boldsymbol{w})]$. The regret at step $t$ is defined as

$$\text{Regret}^t = U(\boldsymbol{y}^*_{\text{true}}; \boldsymbol{w}_{\text{true}}) - U((p^t, h(p^t)); \boldsymbol{w}_{\text{true}}). \tag{24}$$

**Baselines.** We compare PACE with strong baselines from the interactive MOO literature, all of which use GPs to model objective functions (privacy and accuracy in our case). Therefore, we follow the method proposed by Avent et al. (2020) to build surrogates for privacy and accuracy. Details can be found in Appendix D. These baselines employ different acquisition functions for preference learning, but all of them use expected improvement (EI) to pick candidates in optimization stage. 1) **BALD** (Ozaki et al., 2024) uses Bayesian Active Learning by Disagreement to select which pairs to query the decision-maker. 2) **UU** (Astudillo & Frazier, 2020) chooses query pairs randomly based on the utility function. 3) **Pareto** (Ungredda et al., 2022) also presents the Pareto fronts to the decision-maker for preference learning. The original work presents the Pareto front only once and assumes the decision-maker is perfect. We extend this method to be iterative—our interactive setting—and use our user model for inferring preferences, which allows the noisy feedback from decision makers. 4) **qEUBO** (Astudillo et al., 2023) chooses pairs that maximize the expected utility. We set $q = 4$, as proposed by Astudillo et al. (2023), which means 4 solutions are picked to compare. 5) **TS** (González et al., 2017) employs dueling Thompson sampling as one of the dueling acquisition functions for preference learning. 6) **EPIG** (Smith et al., 2023) proposes the expected predictive information gain, an acquisition function that measures information gain in the space of predictions rather than parameters.

To demonstrate the effectiveness of PACE, we conduct experiments across two distinct and representative machine learning scenarios. The ablation studies on the Pareto font and preference learning can be found in Appendix B.

## 5.2 Finding the Preferred Trade-off in Differentially Private Logistic Regression

First, we address a classic privacy-preserving task: train a logistic regression model with DP-SGD on the Adult (Becker & Kohavi, 1996) and Dutch (Van der Laan, 2000) benchmark datasets. We consider HPO for batch size, learning rate, clipping threshold, and epochs and adopt the same search ranges as in Avent et al. (2020), which can be found in Table 1. The range of $\varepsilon$ is set as $[0.01, 0.5]$ and we fix $\delta = 10^{-5}$.

Figure 3 presents the observed model accuracy (blue dots) after HPO as a function of the privacy budget, $\varepsilon$, for the Adult and Dutch datasets. We fit two S-shaped surrogate models to this empirical data: the symmetric

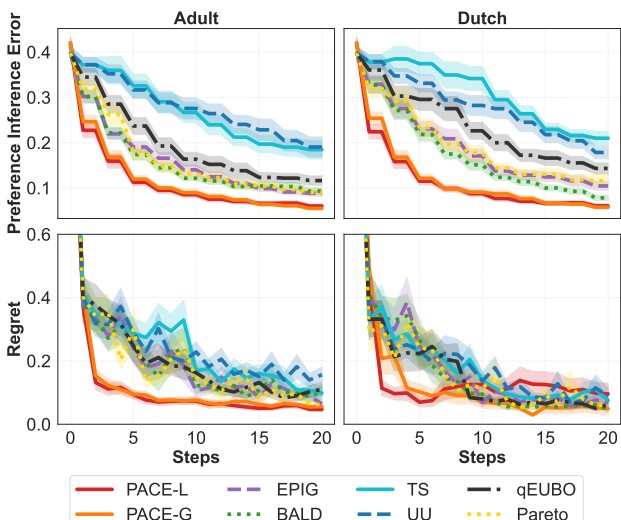

Figure 4: Results of DP logistic models. The top row shows the preference inference errors and the bottom row shows the regrets. PACE-L and PACE-G converge four times faster than baselines.

Logistic function (red line) and the asymmetric Gompertz function (orange line). **Although these trade-off curves do not represent complete S-shapes—they do not reach a lower asymptote even for very small epsilon values, the flexibility of the S-shape functions still yields a good fit.**

Figure 4 illustrates the performance of PACE using Logistic and Gompertz surrogates, PACE-L and PACE-G, against the baselines, evaluated on two key metrics: preference inference error and regret.

The top row shows the preference inference errors. Across both datasets, PACE-L and PACE-G (red and orange lines) achieve lower preference errors than all baselines. This demonstrates their better sample efficiency, as they can learn the decision-maker's preferences more accurately with fewer interactions.

The bottom row shows the regrets, which measures how effectively PACE finds the decision-maker's preferred trade-off. PACE-L and PACE-G converge to lower regret faster than the baselines. As observed from the regret curves, PACE approximately converges using 80 hyperparameter configuration evaluations, while baselines require 200 evaluations to reach a comparable regret level—a roughly 2.5× reduction. On the Dutch dataset, while PACE-L performs well in the initial steps, PACE-G ultimately converges to a lower final regret, suggesting it forms a more accurate long-term model of the Pareto front.

## 5.3 Finding the Preferred Trade-off in Differentially Private Deep Transfer Learning

In this section we conduct more advanced experiments using a Vision Transformer (Dosovitskiy et al., 2021) pretrained on ImageNet-21k (Ridnik et al., 2021) from the PyTorch Image Models library (Wightman, 2019) and fine-tuned on the target task with DP-Adam (Tobaben et al., 2023). We test PACE on four datasets: CIFAR-100 (Krizhevsky & Hinton, 2009), SVHN (Netzer et al., 2011), SUN (Xiao et al., 2010) and PatchCamelyon (Veeling et al., 2018). The details of the training process can be found in Appendix E.

We fix the number of epochs to 40—as it empirically gives strong performance in our experiments—and optimize the learning rate, the batch size, and the clipping threshold. As we expect smaller values of the learning rate and the clipping threshold to give better results, we search for those in log space. The search bounds for these hyperparameters are shown in Table 1.

We also show the two S-shaped models fitting in Figure 5 for four datasets. The range of $\varepsilon$ is set as $[0.01, 16]$ for CIFAR100, SVHN and PatchCamelyon, and $[0.01, 64]$ for Sun397. Both functions provide a good fit to the observed data. The top row in Figure 6 shows that across all datasets, PACE achieves consistently smaller preference errors than baselines. The bottom row indicates that PACE approximately converges using 80 hyperparameter configuration evaluations, while baselines still achieve worse regret after 200 evaluations.

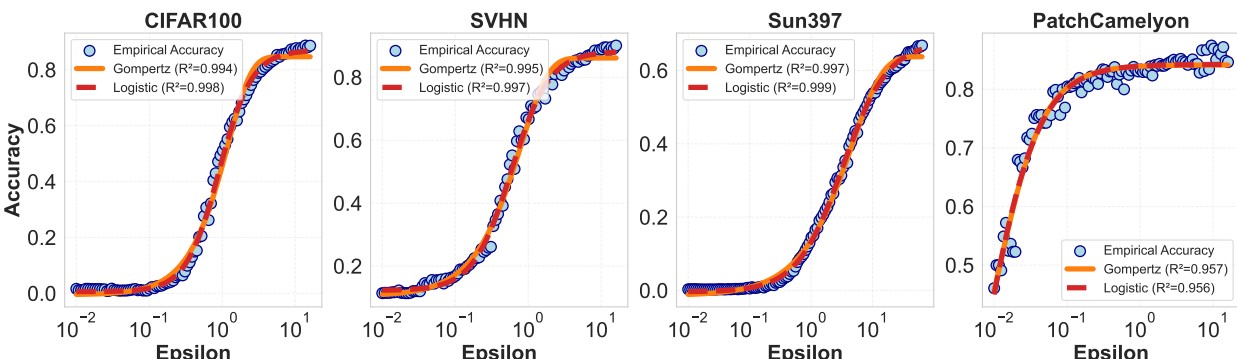

Figure 5: Logistic and Gompertz functions fitted to the empirical trade-off for DP deep transfer learning models with $\delta = 10^{-5}$. Both curves effectively capture the trend of the observed trade-off data.

Our experiments support the use of S-shaped surrogates across the diverse curve shapes observed here. For instance, the trade-off curve for the PatchCamelyon dataset in Figure 5 represents only the steep portion of an S-shaped curve, while the curve for Sun397 does not reach its upper asymptote even with a large privacy budget $\varepsilon = 64$. In both cases, PACE captures the underlying trend well, showcasing its flexibility. This highlights a key advantage of our framework: our results suggest that **it does not require the full S-shaped curve to be present.** Decision-makers can define any specific range of epsilon that is relevant to their application, and PACE can efficiently identify their preferred trade-off within that custom domain.

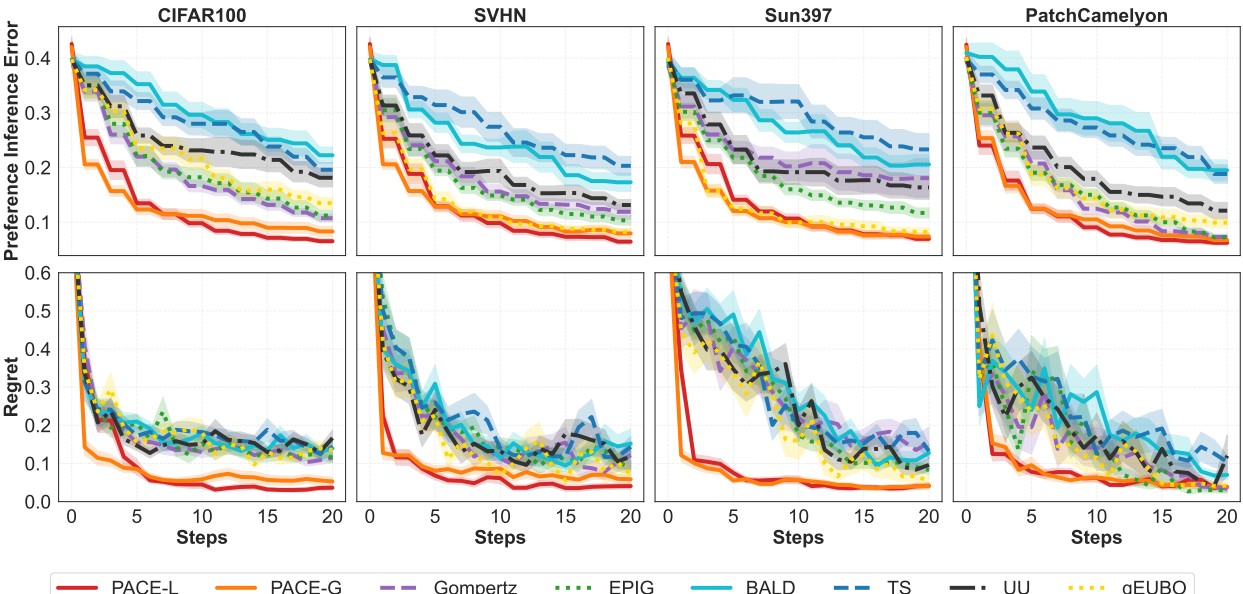

Figure 6: Results of DP deep transfer learning results. The top row shows the preference inference errors and the bottom row shows the regrets. PACE-L and PACE-G converge four times faster to lower preference-learning error and regret than baselines.

## 6   Related Work

**Learning trade-off in DP**   Prior works learn the trade-off in DP using different MOO methods. Avent et al. (2020) formulated the trade-off between $\varepsilon$ and classification error as multi-objective Bayesian optimization and use hypervolume as a metric to perform HPO to get the entire Pareo front. Priyanshu et al. (2022) considered the linear combination of $\varepsilon$ and validation loss and employ three different methods—Bayesian optimization, reinforcement learning and evolutionary approach for HPO. However, Priyanshu et al. (2022) assumed the preference weights are known beforehand, leaving the critical question of how to choose them

unanswered. Moreover, Arcolezi & Gambs (2025) used MOO to address the trade-off in local DP protocols and Ranaweera et al. (2025) explored the trade-off in federated learning setting.

The challenge of selecting an appropriate privacy level is a well-recognized open problem in the practical deployment of DP. Foundational work highlights that the right choice of $\varepsilon$ is not universal but is highly context-dependent, varying between different applications and datasets (Dwork et al., 2019). To address this, one line of research focuses on making $\varepsilon$ more interpretable. These works aim to translate the abstract $\varepsilon$ value into more concrete and understandable measures of privacy risk, helping practitioners make a more informed judgment call (Cummings et al., 2021; Nanayakkara et al., 2023). Another direction, more aligned with our own, focuses on building interactive systems. Nanayakkara et al. (2022) developed an interface to help users visualize the trade-offs between $\varepsilon$, accuracy, and disclosure risk. Nanayakkara et al. (2024) proposed an interactive paradigm for exploratory data analysis which allows a user to start with a small $\varepsilon$ and iteratively increase it if better utility is required. However, it still requires a predefined privacy budget, which is difficult to set without seeing the trade-off curve.

Another perspective on the trade-off is the *accuracy-first* paradigm (Ligett et al., 2017; Wu et al., 2019; Rogers et al., 2023; Räisä et al., 2025). These methods typically frame the problem inversely: given a pre-defined accuracy target, they aim to minimize the privacy budget required to achieve it. While valuable when strict performance requirements exist, this approach faces a cold start problem similar to choosing $\varepsilon$: setting an appropriate accuracy target without prior knowledge of the trade-off landscape is difficult. As highlighted by Cyffers (2025), the limitations of focusing on a single metric are widely documented in the machine learning community. Our work addresses this by enabling the decision-maker to explore the entire trade-off curve before committing to specific targets for either privacy or accuracy.

**Interactive MOO**   Incorporating the decision-makers' preferences into exploration algorithms has been studied in MOO literature (Lin et al., 2022; Ip et al., 2025). Astudillo & Frazier (2020) proposed a multi-attribute BO algorithm with a preference-based expected improvement acquisition function. Ozaki et al. (2024) proposed active learning for preference query in interactive MOO. Ungredda & Branke (2023) considered KG acquisition functions in both preference learning and objectives estimation procedures, and proposed a general framework on when to elicit preferences which we build our work on. However, these methods all use pairwise comparisons to collect decision-makers feedback. While it is easy to implement, each comparison unfortunately provides only a small amount of preference information, and hence the approaches requires a considerable number of queries. Ungredda et al. (2022) proposed to solve this limitation by generating the estimated Pareto front using an evolutionary algorithm and presenting all discrete points on the Pareto font to the decision-maker. However, due to the computational difficulty of generating the estimated front, they only perform a single round of interaction. Practical applications of interactive MOO are attracting growing interest. Giovanelli et al. (2024) studied the trade-off in HPO between accuracy and power consumption.

## 7   Discussion

**Setting the range of the privacy level**   Our framework requires the decision-maker to pre-define a range of interest for the privacy level (e.g., $[\varepsilon_{\min}, \varepsilon_{\max}]$ under the $(\varepsilon, \delta)$-DP with a fixed and small $\delta$). There is no universally "correct" range; the appropriate bounds are context-dependent and are dictated by the specific application's trade-off. For instance, applications involving highly sensitive data, such as analyzing patient records in a healthcare setting, demand stringent privacy guarantees and would thus necessitate a range with a very low upper bound (e.g., $\varepsilon \leq 1$) (Dankar & El Emam, 2012). Conversely, in scenarios where the practical threat model emphasizes robustness against specific, empirically observed attack vectors—such as Membership Inference Attacks (MIA)—a decision-maker may consider larger privacy budgets acceptable in practice. Some empirical studies report that current MIAs may struggle to achieve high success rates even at relatively large $\varepsilon$ values (e.g., $\varepsilon > 8$), suggesting a practical gap between the worst-case theoretical guarantee and the realized risk of current attacks (Nasr et al., 2021; Lowy et al., 2024). In such contexts, a decision-maker might prioritize model accuracy and select a higher $\varepsilon$, accepting a weaker theoretical bound in exchange for preserving critical service quality.

PACE is designed to operate within such a user-defined range, regardless of its specific bounds. The goal of our framework is not to prescribe a universal privacy level, but rather to provide a principled tool that helps a decision-maker efficiently identify their preferred privacy-accuracy trade-off within their own pre-defined spectrum of acceptable options.

**Likelihood for observed accuracies**  In our current framework, we model the observed accuracy using a Gaussian likelihood with a constant variance ($\sigma^2$). This homoscedastic assumption implies that the reliability of our accuracy measurements is the same across the entire range of the privacy level, $p$.

However, it is reasonable to suspect that this assumption could be refined. One could hypothesize a heteroscedastic relationship, where the variance of the observed accuracy is itself a function of $p$. For instance, at very large $p$ values, the large amount of injected noise makes the training process highly unstable, likely leading to a larger variance in the final accuracy across different runs. Conversely, at small $p$ values, the training process is more stable and deterministic, suggesting a smaller variance.

Future work could incorporate this by modeling the variance as a function of the privacy level. Adopting such a heteroscedastic model could lead to a more accurate representation of the true data-generating process and potentially improve the sample efficiency of the Pareto front exploration.

**Target Audience**  The primary audience for our work is data practitioners and researchers who are familiar with interpreting privacy parameters and are responsible for selecting the desired privacy level to deploy the differentially private models. Our aim to provide a sample-efficient method to aid this complex decision-making process. However, we consider the important challenge of how to best communicate the nuances of differential privacy to a non-technical audience to be outside the scope of this paper. This remains an active and valuable area of research, with notable work exploring how to translate formal privacy guarantees into understandable, practical terms for the general public (Cummings et al., 2021; Nanayakkara et al., 2023).

**Limitations**  A limitation of our current framework is that it does not provide a formal end-to-end privacy guarantee for the entire optimization process. Specifically, the intermediate adaptive queries used to learn the trade-off curve involve repeated access to the private dataset. Under a worst-case theoretical model where the decision-maker is considered an adversary, this adaptive interaction would accumulate significant privacy loss. However, PACE is designed for the *trusted curator* threat model common in practical deployments (Avent et al., 2020), where the optimization is an internal process conducted by a trusted team, and only the single, final model is released to the public. Within this scope, PACE provides a principled and sample-efficient way to select the privacy level for that final release. Developing a fully private, end-to-end version of this interactive framework remains a challenging and important direction for future research.

Another limitation is the absence of a user study with real-world practitioners. Our primary goal was to design and rigorously evaluate the algorithmic efficiency of PACE in a controlled, reproducible environment. We therefore followed standard methodology in interactive optimization by employing a simulated user. While this validates the method's performance in a controlled simulated-user setting, it does not evaluate the usability of the interface or the potential impact of cognitive biases. Conducting a formal user study to validate the framework's practical utility with human decision-makers is a critical direction for future work.

Lastly, our theoretical analysis is derived for logistic regression, and our empirical evaluation focuses on gradient-based models with linear classification heads. We demonstrate that the S-shaped relationship also works for cross-entropy loss in Appendix G. Whether the same S-shaped approximation extends to fundamentally different learning paradigms, such as decision trees or random forests, remains an open question and is an interesting direction for future work.

## 8    Conclusion

In this work, we proposed a more sample-efficient approach for finding the preferred trade-off between privacy and accuracy with respect to a decision-maker's preference in differential privacy. To motivate our modeling choice, we first provided a theoretical characterization of the trade-off in the specific setting of logistic regression under $\varepsilon$-DP, and complemented this analysis with empirical evidence across a broader range of

differentially private learning pipelines. Together, these results support the use of S-shaped functions as effective surrogates for approximating the observed trade-off curves. Building on this insight, and leveraging the observation that hyperparameter tuning at a fixed privacy level yields approximate Pareto-front solutions, we developed a Bayesian method for explicitly learning and modeling the Pareto front. Additionally, using this probabilistic model, we proposed a sample-efficient interaction scheme to infer the preferences over the privacy-accuracy trade-off. Putting these components together, we demonstrated that PACE outperforms the compared baselines on our regret and preference-inference metrics. Importantly, our framework does not require observing the full trade-off curve or its asymptotic regimes. Instead, it leverages S-shaped functions as flexible local surrogates that can approximate diverse trends with good sample efficiency.

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

# A  Experiments Configurations

Here we list the prior distributions for parameters of Logistic and Gompertz functions in all experiments.

Table 2: Priors distribution for parameters of Logistic and Gompertz functions.

| Parameter | Logistic | Gompertz |
|---|---|---|
| $L$ | Beta$(40, 2)$ | Uniform$(0.8, 4)$ |
| $k$ | LogNormal$(\log(10), 0.2)$ | Uniform$(10, 100)$ |
| $c$ | Beta$(2, 2)$ | Uniform$(1, 10)$ |
| $b$ | Normal$(0, 0.1)$ | Uniform$(0.8, 1.1)$ |
| $\sigma$ | Gamma$(0.5, 0.1)$ | Gamma$(0.5, 0.1)$ |

# B  Ablation Study

In this section, we investigate the individual components that make up PACE and show the performance.

## B.1  Pareto Front Estimation

This experiment investigates the sample efficiency of PACE—modeling the Pareto front directly—against the standard baseline approach of modeling the objective functions (recall Section 2.2). While our framework supports various S-shaped curves to model the front, we found that the Gompertz and Logistic functions performed similarly across various datasets. Therefore, for clarity and illustration, we use the Logistic function in the following experiments. We assume the true preferences weights are known and compare the regret (Equation 24) on finding the preferred trade-off of CIFAR100. We compare PACE to other three baselines: 1) PACE-L, which selects values for constraint optimization with KG. 2) **Logistic-random**: ablation approach which randomly selects constraints to optimize with. We add two (traditional) approaches that infer the Pareto front from estimating the objective functions: 3) **GP-KG**: select candidates ($\boldsymbol{\theta}$) from the hyperparameter space that optimize KG. 4) **GP-random**: randomly select candidates from the hyperparameter space.

Results in the right pane of Figure 7 shows the comparisons. Modeling the Pareto front requires clearly fewer samples than modeling the objectives. The regret obtained when modeling the Pareto front with a Logistic function (using either random sampling (green) or our acquisition function (red)) is consistently lower than when modeling the objectives directly. Note that while random sampling performs better in the initial steps, it converges to worse regret, whereas PACE-L continues to reduce regret further. This shows that random sampling cannot identify the exact optimal trade-off because it does not exploit preference information.

## B.2  Preference Learning

Here, we focus on the sample efficiency of preference learning given the true Pareto front, which is assumed to be known. We measure the preference inference error (Equation 23) on $\boldsymbol{w}$, given feedback from the decision-maker. In this task, we compare PACE to the other three methods: (1) PACE-L, which selects the next curve with KG. (2) **Random curve**: querying random hypothetical Pareto fronts, (3) **Pairs with KG**: pairwise comparisons picked with KG, and (4) **Random pairs**: random pairwise comparisons.

Results in the left pane of Figure 7 confirm our expectations: random pairwise comparisons are the least informative, while pairs picked with KG perform similarly to random hypothetical Pareto fronts. Optimizing for informative fronts—PACE-L—leads to clearly the lowest inference error.

## B.3  Visualization of Utility Function

In this section, we visualize the behavior of the Chebyshev utility function in comparison to linear utility functions when applied to the S-shaped privacy-accuracy Pareto front to support our choice. We have $w_1 + w_2 = 1$ in all the following utility functions and define $p = -\log \varepsilon$ as previously stated.

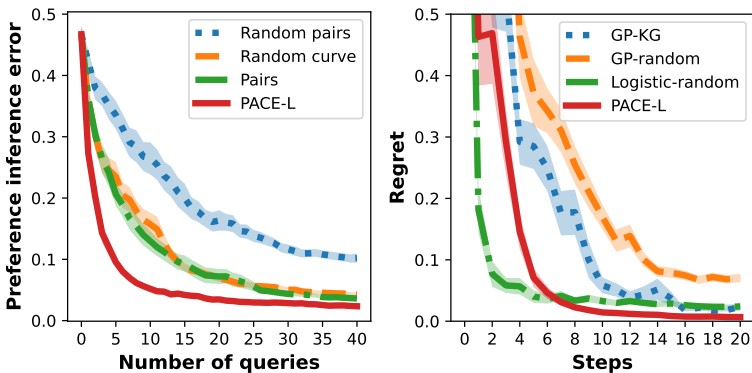

Figure 7: Results of ablation studies. left: ablation study of different interaction mechanisms, where PACE-L has the smallest inference errors. Right: ablation study of different methods to find the most preferred trade-off, where PACE-L converges to a smaller regret.

$$U_{\text{Chebyshev}}(p, \alpha; \boldsymbol{w}) = \min\left(\frac{1}{w_1}\frac{p - p_{\min}}{p_{\max} - p_{\min}}, \frac{1}{w_2}\frac{\alpha - \alpha_{\min}}{\alpha_{\max} - \alpha_{\min}}\right). \tag{25}$$

$$U_{\text{Linear}}(p, \alpha; \boldsymbol{w}) = w_1 * \frac{p - p_{\min}}{p_{\max} - p_{\min}} + w_2 * \frac{\alpha - \alpha_{\min}}{\alpha_{\max} - \alpha_{\min}}. \tag{26}$$

In linear utility functions, it is possible that: 1) there exists a Pareto optimal solution that cannot be identified by any weight parameters, and 2) only one objective function with the highest weight is exclusively optimized (Ozaki et al., 2024). The left panel of Figure 8 illustrates this failure on the S-shaped Pareto front. This problem is particularly acute for our S-shaped curve as the linear model fails to identify any of the nuanced, balanced trade-offs in the crucial middle region of the curve. The left panel of Figure 8 shows that for nearly all preference weights, the maximum utility is achieved at one of the two endpoints of the Pareto front.

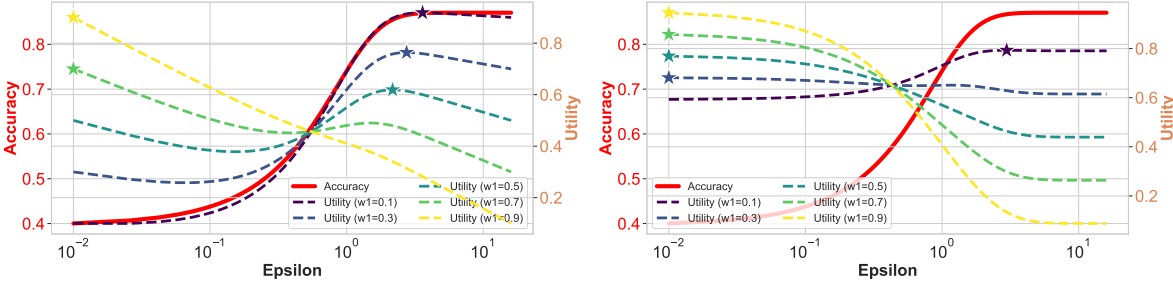

Figure 8: Visualization of linear utility functions (Equation (26) in the left panel and Equation (27) in the right panel) for varying preference weights. The red solid curve presents the privacy-accuracy trade-off. Each star point marks the optimal solution that maximizes the utility for its corresponding weight, which fails to identify trade-offs in the crucial middle region of the curve.

We also visualize the linear utility proposed in Priyanshu et al. (2022) in the right panel of Figure 8. They consider the weighted linear combination of validation loss and $\varepsilon$ as the utility function for HPO. As we explore the S-shaped trade-off curve between $\varepsilon$ and $\alpha$, we follow the idea in Avent et al. (2020) and take classification error $(= 1 - \alpha)$ as the metric for model performance. Then we have the following utility function according to Priyanshu et al. (2022):

$$U_{\text{Linear}(Priyanshu\ et\ al.,\ 2022)}(\varepsilon, \alpha; \boldsymbol{w}) = w_1 * e^{-\varepsilon} + w_2 * e^{\alpha - 1}. \tag{27}$$

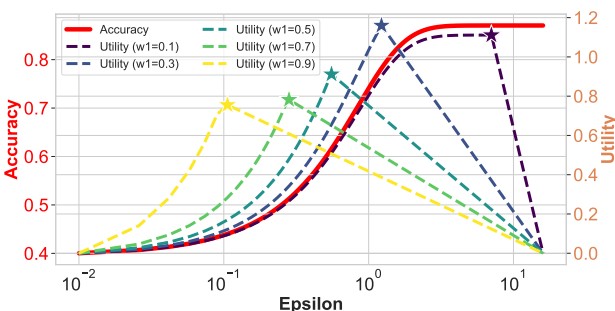

Figure 9: Visualization of Chebyshev utility function for varying preference weights. The red solid curve presents the privacy-accuracy trade-off. Each star point marks the optimal solution that maximizes the utility for its corresponding weight, which can successfully identify a diverse range of intermediate trade-offs.

The right panel of Figure 8 shows the same issue with the left panel—for nearly all preference weights, the maximum utility is achieved at one of the two endpoints of the Pareto front.

In stark contrast, Figure 9 demonstrates the suitability of the Chebyshev utility function for our framework. We can see that as the decision-maker's preference weight changes, the peak of the utility curve shifts smoothly across the Pareto front. The star points, which mark the optimal trade-off for each weight, clearly show that the Chebyshev utility can identify a diverse range of intermediate trade-offs. Therefore, a decision-maker can flexibly select any trade-off on the S-shaped Pareto front, and PACE can infer the corresponding preference weights that would make that choice optimal. This creates a robust mapping from a decision-maker's action back to their latent preferences.

## C  Sensitivity Analysis

### C.1  Sensitivity Analysis of $T$ in User Modeling

Our proposed user modeling Equation (7), which is shown below again,

$$P(\boldsymbol{y^*} \mid \boldsymbol{\beta}, \boldsymbol{w}) = \frac{\exp(U(\boldsymbol{y^*}; \boldsymbol{w})/T)}{\sum_j^q \exp(U(\boldsymbol{y}_j; \boldsymbol{w})/T)},$$

where $T$ is the rationality coefficient. While we take it as known in the preference learning procedure, inferring it along with other parameters is feasible (Yamagata et al., 2024; Astudillo et al., 2023). Here we show the sensitivity analysis on CIFAR100 to show PACE is not sensitive to the inferred $T$.

We assume the Pareto front is known and simulate the decision-maker's action with $T_{\text{true}} = 0.2$ and use $T = 0.1, 0.2, 0.3$ in our user model to learn preferences. We see from Figure 10 that there is no significant difference to use different $T$ in PACE.

### C.2  Sensitivity Analysis of $q$ in User Modeling

As discussed in Section 4.2, we assume the decision-maker discretizes the trade-off curve to pick the most preferred one. In all experiments, we set $q = 100$. Here we present the sensitivity analysis (Figure 11) on Adult to show that PACE is robust to the choice of $q = 50, 80, 100$.

## D  Baselines Implementation

When employing GP to model surrogates of privacy and accuracy in baselines, we use the same Matern kernel as in Avent et al. (2020): $k_M^{5/2}(\boldsymbol{x}, \boldsymbol{x'}) = (1 + \frac{\sqrt{5}|\boldsymbol{x}-\boldsymbol{x'}|}{l} + \frac{5(\boldsymbol{x}-\boldsymbol{x'})^2}{3l^2}) \exp(-\frac{\sqrt{5}|\boldsymbol{x}-\boldsymbol{x'}|}{l})$, where $l$ is the length-scale parameter to be optimized during inference. The number of initial evaluations are 20, and every step 20 evaluations are sampled based on EI acquisition function.

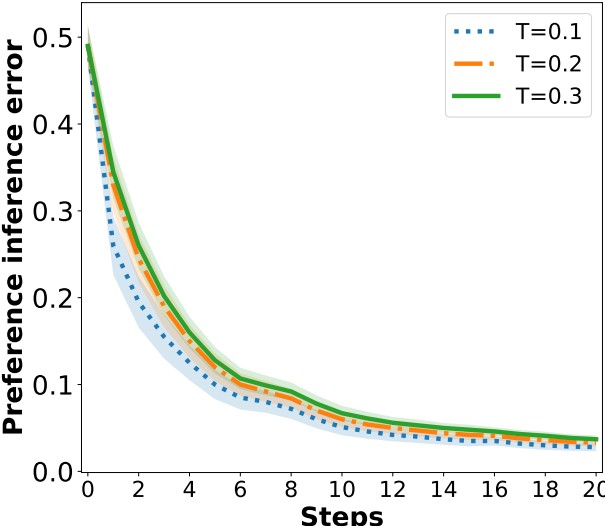

Figure 10: Preference inference errors for different $T$ in the user model on CIFAR100. There is no significant difference between different values.

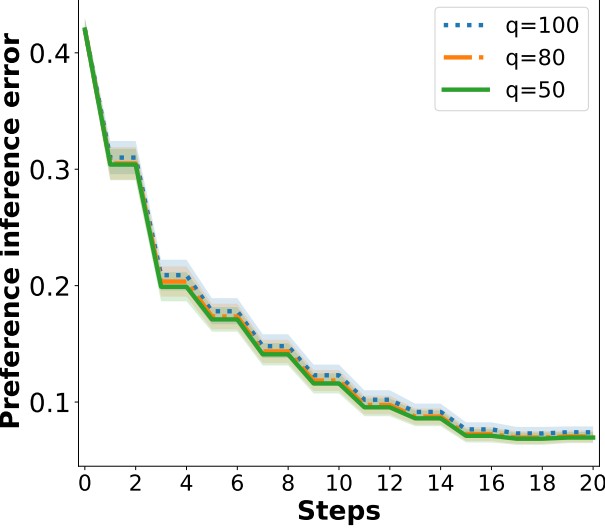

Figure 11: Preference inference errors for different $q$ in the user model on Adult. There is no significant difference between different values.

## E    Details of Training in the Differentially Private Deep Transfer Learning

To facilitate hyperparameter tuning, we decouple the effect of the learning rate and the clipping threshold enabling the learning rate to absorb a fraction of the clipping threshold, as introduced by De et al. (2022). To enable faster training without compromising accuracy, we leverage DP-FiLM, as introduced by Tobaben et al. (2023). In DP-FiLM, we freeze all model parameters except for the scale and bias of the normalization layers and the final classification layer. Following the DP-FiLM method, we initialize the classification layer weights to zero and make them trainable, resulting in 0.13-0.40% of trainable parameters depending on the

classification head size. The pretrained model expects $224 \times 224$ input images, so we resize images accordingly. Training is performed in distributed mode across 4 AMD Radeon Instinct MI250X GPUs.

## F  Computation of Acquisition Functions

For efficient approximation of the posteriors, we employ Monte-Carlo estimates with importance sampling (Elvira & Martino, 2022). We derive this for preference learning; the same approach is also applied to Pareto front estimation.

Assume $\{\boldsymbol{y}_m^*, \boldsymbol{\beta}_m\}_{m=1}^M$ have been observed. When a new observation $(\boldsymbol{y}_{M+1}^*, \boldsymbol{\beta}_{M+1})$ comes, the target posterior is

$$
P(\boldsymbol{w} \mid \{\boldsymbol{y}_m^*, \boldsymbol{\beta}_m\}_{m=1}^M, (\boldsymbol{y}_{M+1}^*, \boldsymbol{\beta}_{M+1})) \propto
$$
$$
P(\boldsymbol{y}_{M+1}^* \mid \boldsymbol{\beta}_{M+1}, \boldsymbol{w}) \times \prod_{m=1}^M P(\boldsymbol{y}_m^* \mid \boldsymbol{\beta}_m, \boldsymbol{w}) \times P(\boldsymbol{w}) \propto
$$
$$
P(\boldsymbol{y}_{M+1}^* \mid \boldsymbol{\beta}_{M+1}, \boldsymbol{w}) \times P(\boldsymbol{w} \mid \{\boldsymbol{y}_m^*, \boldsymbol{\beta}_m\}_{m=1}^M). \tag{28}
$$

Importance sampling constitutes two steps:

1. **Sampling:** $n_{\boldsymbol{w}}$ samples are simulated from
$$
\boldsymbol{w}_r \sim P(\boldsymbol{w} \mid \{\boldsymbol{y}_m^*, \boldsymbol{\beta}_m\}_{m=1}^M, \; r = 1, \ldots, n_{\boldsymbol{w}}.
$$

2. **Weighting:** Each sample receives an associated importance weight given by
$$
p_r = \frac{P(\boldsymbol{w}_r \mid \{\boldsymbol{y}_m^*, \boldsymbol{\beta}_m\}_{m=1}^M, (\boldsymbol{y}_{M+1}^*, \boldsymbol{\beta}_{M+1}))}{P(\boldsymbol{w}_r \mid \{\boldsymbol{y}_m^*, \boldsymbol{\beta}_m\}_{m=1}^M)}
$$
$$
= P(\boldsymbol{y}_{M+1}^* \mid \boldsymbol{\beta}_{M+1}, \boldsymbol{w}_r), \tag{29}
$$

which is exactly the likelihood of the new observations.

Similarly, in trade-off curve inference in DP case, the importance weight is
$$
p_s \approx \mathcal{N}(a_n \mid h(p_n; L, k, b, c), \sigma^2) \times d_n,
$$

where $d_n$ is a small interval.

The importance weights describe how representative the simulated samples are. The set of $n_{\boldsymbol{w}}$ weighted samples can be used to approximate the target posterior. We sample from the prior distribution of $\boldsymbol{w}$ and update weights as new observations arrive. Algorithm 2 details this importance sampling approach to KG computation.

## G  Pareto Front Modeling with Cross-Entropy Loss

Throughout the main paper, we model the trade-off between the privacy level and classification accuracy, as accuracy is the metric used in the downstream preference learning task. Now we examine whether the observed S-shaped relationship also holds for other performance measures. To investigate this, we repeat the curve-fitting experiment from Section 5.2 using the cross-entropy loss instead of accuracy as the performance metric. The experimental setup, hyperparameter optimization procedure, and privacy budgets remain unchanged.

Figure 12 shows the resulting trade-off curves for the Adult and Dutch datasets. Both the Gompertz and Logistic functions continue to provide an excellent fit to the empirical observations, achieving comparable $R^2$ values to those obtained with accuracy. These results suggest that the proposed S-shaped surrogate is not specific to classification accuracy, but can also effectively model the relationship between privacy and cross-entropy loss. While we only consider cross-entropy loss here, extending this empirical analysis to other performance metrics such as F1-score is an interesting direction for future work.

---

**Algorithm 2** Simulation and importance-sampling based computation of KG in preference learning.

---

**Input:** **Candidate curves, number of simulations Num, posterior of $\boldsymbol{w}$.**
**for** All candidate curves **do**
   **for** $\mathrm{sim} = 1, \ldots, \mathrm{Num}$ **do**
      Generate a simulated action based on the posterior of $\boldsymbol{w}$.
      Update the posterior of $\boldsymbol{w}$ based on the simulation using (29).
      $\Delta^{\mathrm{sim}} = U^*_{M+1,N} - U^*_{M,N}$.
   **end for**
   Estimate KG by $\frac{1}{\mathrm{Num}} \sum_{\mathrm{sim}=1}^{\mathrm{Num}} \Delta^{\mathrm{sim}}$.
**end for**
**Output:** The curve with the highest KG value.

---

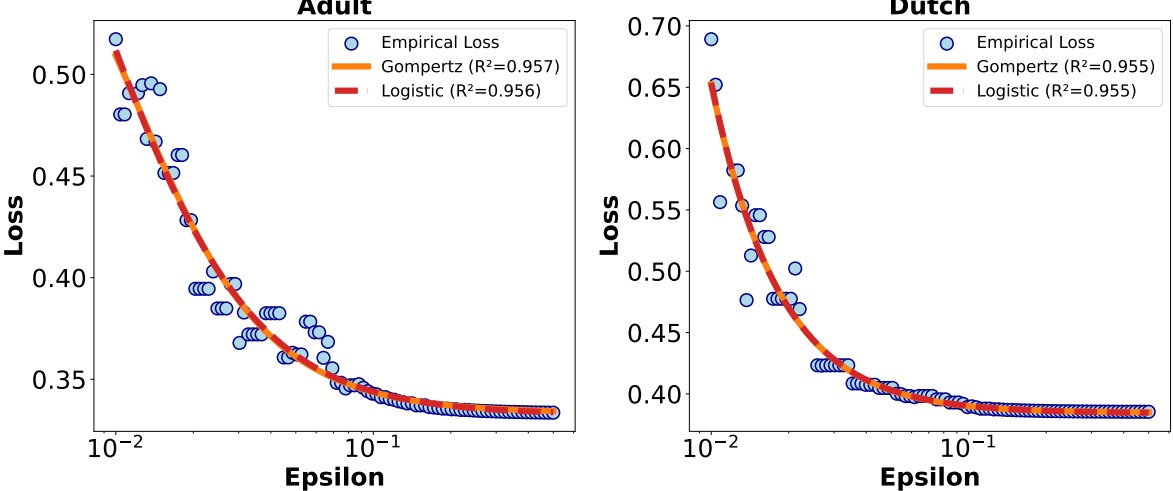

Figure 12: Logistic and Gompertz functions fitted to the empirical privacy-loss trade-off for DP logistic regression on the Adult and Dutch datasets with $\delta = 10^{-5}$. Both curves effectively capture the trend of the observed trade-off data.

