# OpenReview forum: "An Interactive Framework for Finding the Optimal Trade-off in Differential Privacy"
_TMLR — Under review for TMLR_

### Review · Reviewer_tgyP · 2026-06-02

**Summary Of Contributions:**

The paper „An Interactive Framework for Finding the Optimal Trade-off in Differential Privacy” presents the PACE method, which at its core assumes that the accuracy-privacy trade-off can be modelled by assuming privacy is a constraint (i.e., must be at a certain level), thereby treating the multi-objective problem as a single objective problem (accuracy) with a constraint (privacy). Using this, they estimate S-shaped curves that present the optimal accuracy-privacy trade-off to select the best utility (of the trade-off) while still finding Pareto-optimal solutions with a low budget of user interactions. Simulated data shows that this works well both with a logistic regression model (2 tasks), as well as with a vision transformer (4 tasks).

**Audience:**

Yes

**Audience Explanation:**

The paper presents a new method with a good idea at its core to simply a multi-objective problem to a single objective problem and then exploits this nicely to make the solution-finding on the pareto-front more efficient.

**Claims And Evidence:**

Yes

**Claims Explanation:**

I enjoyed reading the paper. The method is clear, the reframing of the problem into a single-objective problem with a constraint makes sense to me and can be generalized to all two-objective problems. In this case, there might also be a preference for accuracy (must achieve a certain level to be usable) and then try to determine the most private model instead. The only concern with this is whether the S-shape assumption still holds. Nevertheless, this is an interesting opportunity for future work that could be mentioned. A strong point is that the experiments are not limited to logistic regression, which is used for the analysis, but also use modern deep learning models. However, I note that the prediction head is still of the deep learning model is still a logistic regression model (linear layer with softmax/cross-entropy loss). Whether the S-shape is also there for models with a completely different prediction head that uses loss that is not directly linked to logistic regression (e.g., a Random forest) or something that is likely not continuous (e.g., a Decision Tree) is unclear. This should be mentioned as limitation.

The paper itself is (mostly) well-structured and easy to read. I was just a bit confused by the location of the related work, between the method and the experiments. This should either be moved up to be at the beginning (where most of this is already discussed anyways) or to a place after the results.

**Requested Changes:**

Suggested changes:
- Page 6: Uses the P and A without double stroke in Section 4.
- Page 7: Z is undefined in the paper. This should be added to improve the readability.
- It is a bit unclear why Gompertz and Logistic are both used. Their shapes are almost identical and there is no visible difference between their performance. The paper should be sharpened a bit by only using one function (the simpler Logistic function?), once it was established that there is no notable difference.
- The related work should be moved so it is not between the method and the experiments.
- The limitations should specify that all models use logistic regression as prediction head and that it is unclear if the same S-shape would also be observed with models that use prediction heads with different structures like random forests or decision trees.

---

> ### Author Response · Authors · 2026-06-30
> **Response to reviewer tgyP**
>
> Thanks for your review. We have uploaded a revision and highlighted changes to the original submission in blue.
>
> **Requrested changes:**
>
> > 1. Presentation issues.
>
> Thanks for catching these. We have fixed them in the revised version.
>
> > 2. It is a bit unclear why Gompertz and Logistic are both used. Their shapes are almost identical and there is no visible difference between their performance. The paper should be sharpened a bit by only using one function (the simpler Logistic function?), once it was established that there is no notable difference.
>
> Gompertz function comes from the theoretical derivation of the trade-off curve for regularized logistic regression. Logistic function is introduced as a more general and widely used surrogate. The experiments show that both work well. The point is not to compare Gompertz vs Logistic. The point is to demonstrate that the framework is **not tied to a specific S-shaped parameterization**. We clarified this discussion on Page 9.
>
> > 3. The related work should be moved so it is not between the method and the experiments.
>
> We have moved it to Section 6, before the discussion section.
>
> > 4. The limitations should specify that all models use logistic regression as prediction head and that it is unclear if the same S-shape would also be observed with models that use prediction heads with different structures like random forests or decision trees.
>
> Thanks for bringing up this good point. We added this limitation to the Section 7 (discussion) on Page 16.

---

> > ### Comment · Reviewer_tgyP · 2026-07-01
> >
> > Thank you for updating your paper. I checked and am satisfied with these changes.

---

### Review · Reviewer_2gn4 · 2026-06-08

**Summary Of Contributions:**

## Summary:
This paper presents **Privacy-Accuracy Curve Elicitation (PACE)** as new technique for approximating a user-preferred trade-off between accuracy and differential privacy on their Pareto front. In the context of the $(\\epsilon, \\delta)$ differential privacy framework,  PACE addresses two main challenges:

* **Approximating the Pareto front:** Instead of finding the Pareto front with a basic grid approach over the privacy budget $\\epsilon$ or any surrogate model, theoretical and empirical insights show that an S-shaped function (i.e., logistic or Gompertz) is a good proxy for the Pareto front.
* **Characterizing the user preferences:** Approximating the Pareto front is not sufficient since the question which solution on this front is best depends on the user's preferences. Instead of using common tools from multi-objective optimization, e.g., pairwise queries or a reference point, the currently approximated S-shaped Pareto front is presented to the user for picking their preferred solution on this front.

To minimize the costs for the hyperparameter optimization (to approximate the Pareto front) and user effort (to characterize their preferences), a **knowledge-gradient is employed as acquisition function**, which is maximized to find the most informative S-shaped function to present to the user preference and the most informative privacy constraint to optimize accuracy for. An empirical evaluation demonstrates a low user preference error and utility regret of PACE across 2 tabular datasets with logistic regression as classifier as well as for 4 image datasets with a deep neural network as classifier.

## Strengths:
- PACE is a **unifying framework for differential privacy** because it jointly combines approximating the Pareto front of accuracy and privacy budget with learning user preferences in a cost-efficient (computational resources and user queries) way.
- The **empirical evaluation is well designed** because it includes a reasonable selection of baselines, covers tabular and image datasets, two types of classification models, and includes important ablations confirming the rationality of the design choices involved by PACE.
- The **paper is clearly structured and well-written**. For example, the research gap in the form of the aforementioned two challenges is clearly motivated and its solution is well communicated.
- The discussion is **transparent about limitations**, e.g., assumption of a trusted curation thread model and no experiments with real human users.

## Weaknesses:
- **The claim of the title is too strong** because there is no theoretical guarantee that the framework is able to find the optimal trade-off.
- The S-shaped function surrogate models seem to be **only applicable to accuracy**, while more general surrogate models could be used for other model performance measures.
- The **theoretical insights seem to be limited** to a special logistic regression model, i.e., only binary classification with a uniform data generating distribution.
- **No code has been provided for review**. Moreover, there seems to be no indication (e.g., placeholder comment or supplementary material) that the code to reproduce the empirical evaluation will be made publicly available.

**Additional Comments:**

**Questions:**
- Does the distribution $P(\\boldsymbol{\\beta} \\mid ...)$ in Eq. (19) refer to the distribution in Eq. (18) marginalized over $\\sigma$?
- How or where are the boolean flags "Perform Preference Elicitation" and "Perform Pareto Front Modeling" in Algorithm 1 defined/set?
- When transferring the ($\\epsilon$, $\\delta$) differential privacy into a maximization problem, you use $p=-\\log(\\epsilon)$. Is using a fixed value for $\\delta$ sufficient in this context?

**Minor Editorial Comments:**
- The abstract on OpenReview contains unresolved LaTeX code.
- Section 4 (second paragraph): "... whose exact solution $(p, h(p))$ ~~that~~ lies on the Pareto front."
- Eq. (20): A closing bracket ")" is missing.
- Algorithm 1 and 2: "," is missing in front of "NUM" in the for-loop call.
- Algorithm 2: Input definition is missing.
- Eq. (29): There is an unresolved bracket "(".
- Section 2.3 (first paragraph): Missing capitalization in "... the first in section 2.2, ...".
- Resolve minor consistency issues in the references, e.g., inconsistent writing of venues.

**Audience:**

Yes

**Audience Explanation:**

Differential privacy is an important and broad topic in machine learning research. Moreover, the paper makes clear advancements in this context, which will be **of interest for both, researchers and practitioners**.

**Claims And Evidence:**

Yes

**Claims Explanation:**

Overall, I think that the **paper matches its most important claims** by addressing the two major challenges through PACE that involves reasonable design choices supported by theoretical and empirical insights. Moreover, the discussion clearly points out that PACE does not fully resolve all issues around finding the optimal trade-off.

**Requested Changes:**

It would be great to **address the aforementioned weakness and resolve the additional comments at the end** (i.e., questions and minor editorial issues). Regarding the weaknesses, I would make the following suggestions:

- Adjust the title to not imply that PACE is guaranteed to find the optimal trade-off in differential privacy.
- Add a brief discussion or even better an empirical study to show what happens if we change the model performance measure, e.g., cross-entropy loss or F1-score.
- Communicate earlier (e.g., before the actual derivation) that the theoretical insights are limited to a special regularized logistic regression model.
- Either provide code (e.g., via an anonymous link or supplementary material) or at least indicate whether code will be made publicly available.

**Note:** If I've misunderstood anything, I'll of course adjust the changes I've suggested. I look forward to a productive discussion!

---

> ### Author Response · Authors · 2026-06-30
> **Response to reviewer 2gn4**
>
> Thanks for your review. We have uploaded a revision and highlighted changes to the original submission in blue.
>
> **Weakness and required changes:**
>
> > 1. Adjust the title to not imply that PACE is guaranteed to find the optimal trade-off in differential privacy.
>
> Our original title is "An Interactive Framework for Finding the ***Optimal*** Trade-off in Differential Privacy". Since the trade-off is optimal with respect to user preferences, we revised the title to "An Interactive Framework for Finding the ***Preferred*** Trade-off in Differential Privacy”.
>
> > 2. Add a brief discussion or even better an empirical study to show what happens if we change the model performance measure, e.g., cross-entropy loss or F1-score.
>
> Thank you for the suggestion. We have added an empirical study in Appendix G on Page 27 fitting the S-shaped relationship between cross-entropy loss and privacy. The results show that both Gompertz and Logistic functions continue to fit the empirical trade-off curves well, indicating that the proposed S-shaped surrogate is not specific to the accuracy metric.
>
> > 3. Communicate earlier (e.g., before the actual derivation) that the theoretical insights are limited to a special regularized logistic regression model.
>
> We emphasized in the abstract: “*to efficiently model this trade-off, we theoretically derive the trade-off shape for regularized logistic regression, revealing a characteristic S-curve.*” and “*...We empirically demonstrate its effectiveness across studied DP settings.*” In the introduction, we clarified this point again.
>
> > 4. Either provide code (e.g., via an anonymous link or supplementary material) or at least indicate whether code will be made publicly available.
>
> The code can be found at https://anonymous.4open.science/r/PACE-2DFD.
>
> **Questions:**
>
> > 1. Does the distribution $P(\beta|...)$ in Eq. (19) refer to the distribution in Eq. (18) marginalized over $\sigma$?
>
> Yes. The distribution $P(\beta|...)$ in Eq. (19) refers to the distribution in Eq. (18) marginalized over $\sigma$. We clarified it in the revised version on Page 10.
>
> > 2. How or where are the boolean flags "Perform Preference Elicitation" and "Perform Pareto Front Modeling" in Algorithm 1 defined/set?
>
> As described on Page 10, our default implementation alternates between one Pareto front evaluation and one preference elicitation step. The Boolean flags in Algorithm 1 (Interactive algorithm to identify the most preferred trade-off) indicate which of these two actions is performed at each iteration.
>
> > 3. When transferring the $(\varepsilon, \delta)$-differential privacy into a maximization problem, you use $p=-\log(\varepsilon)$. Is using a fixed value for $\delta$ sufficient in this context?
>
> Yes, when we transform $p=-\log(\varepsilon)$, we fix $\delta$, which is the common setting in practice, and $\varepsilon$ is the main privacy budget varied to study the trade-off.
>
> **Minor editor comments:**
>
> Thanks! We have fixed them in the revised manuscript.

---

> > ### Comment · Reviewer_2gn4 · 2026-07-07
> >
> > Many thanks for the reply, the update of the manuscript, and providing access to the codebase. After inspection, I'm contentwise mostly satisfied with the current state of the submission. However, there are (still) a few editorial issues that should be resolved:
> >
> > * **References:** There are still inconsistent writings of venues, e.g., *Advances in Neural Information Processing Systems* vs. *Advances in neural information processing systems*, *Proceedings of the 23rd International Conference on Artificial Intelligence and Statistics* vs. *International Conference on Artificial Intelligence and Statistics*. Further, the names of some authors are abbreviated e.g., *R Duncan Luce*, while others have their full first name. In this context, name abbreviations are partially ended with and without a dot. Some references also have issues with the author names, e.g., *Adam* is listed as author in *Diederik Kinga, Jimmy Ba Adam, et al. A method for stochastic optimization. In International conference on learning representations, 2015.* These are just examples and I would ask to make sure that all references are correct and consistent.
> > * **Codebase:** The *requirements.txt* does not contain information about the concrete versions of the required Python packages. It would be good to add this missing information or even upload an *environment.yml* file.
> > * **Figures:** I am uncertain whether all figures are high-quality vector graphics, e.g., the graphs in Figure 12 seem to be only images which get blurry when zooming in. Moreover, I would propose to align the drawing of coordinate systems. For example, Figure 10 and 11 differ in their ticks of the *x*- and *y*-axes. These are just examples and I would ask to make all figures vector graphics with a consistent plotting style, if possible.

---

> > > ### Author Response · Authors · 2026-07-08
> > > **Response to the follow-up comments**
> > >
> > > Thank you for the careful reading and detailed suggestions. We have updated the manuscript and codebase to address these editorial issues.
> > >
> > > 1. **References:** We revised the bibliography to make venue names correct and consistent, and we standardized author names by spelling them out consistently.
> > >
> > > 2. **Codebase:** We updated *requirements.txt* to include concrete package versions. We also added an *environment.yml* file so that users can create a pinned Conda environment. The code can be found at https://anonymous.4open.science/r/PACE-2DFD.
> > >
> > > 3. **Figures:** We replaced all figures as vector graphics and improved the consistency of the plotting style across. In particular, we aligned the axis and tick styles in Figures 10 and 11.
> > >
> > > We hope these changes address the editorial concerns.

---

### Review · Reviewer_5xT9 · 2026-06-23

**Summary Of Contributions:**

The paper presents a framework for elicitation of the privacy-utility Pareto frontier using a Bayesian framework. The paper derives the optimal curve for a simple logistic model (with output perturbation DP; not DP-SGD) and argues for an empirically observed similar S curve for significantly larger DP-SGD models.

**Additional Comments:**

The paper could use a proof-reading pass. Some grammar and presentation issues I noticed:

- in the intro, "the" problem structure
- PACE-L and PACE-G are never introduced before their first use in the main matter

**Audience:**

Yes

**Audience Explanation:**

I think facilitating finding the correct operating point on a Pareto frontier is an important topic. The paper makes contributions towards that goal, but as noted I think the target audience is not properly chosen.

**Broader Impact Concerns:**

Not applicable.

**Claims And Evidence:**

No

**Claims Explanation:**

The presentation is not very convincing. There are key issues in terms of accuracy, presentation, novelty and usability. I will detail each with examples which I hope would clarify the required changes.

**Accuracy.** I found some inaccurate claims in the paper. Perhaps the most important one is the "fact that the privacy level in DP naturally serves as a direct optimization constraint" This is simply not true. A privacy budget is at best an upper bound. There is ample empirical and theoretical evidence that these bounds can be quite loose in private training. Assuming that we can tightly optimize accuracy under a privacy constraint is therefore generally not correct. Furthermore, I also believe this characterization is at odds with the empirical section where it is clear that many models are trained with various privacy bounds to achieve better accuracy.

Also, I think both $\beta$ and $w$ decide the Pareto front. The interaction between the two is unclear. The method essentially assume independence between these two factors; which I do not necessarily agree with.

Finally, I do not really follow the theoretical jump from a "decision release mechanism" of Chaudhuri & Monteleoni (2008) in Page 7 to the "gradient release mechanism" of DP-SGD in the later sections of the paper. These mechanisms are not the same. I do not necessarily refute the empirical observations (although experiments on `PatchCamelyon` dataset shows clear divergences from the supposed S-curve); but I find this a logical leap that should stand out to a privacy practitioner.

**Presentation.** The paper has consistency issues. I found claims that contradict each other. Chief among them is the claim that HPO process does not consume privacy budget on Page 11 whereas on Page 4 authors cite Steinke and Papernot 2022 and other work that precisely model this privacy leakage. I was also not at all convinced with the "threat model" of a "trusted curator" that the paper cites at various points in the paper but never truly justifies.    In fact, it is even less clear whose privacy is being protected in light of a trusted curator who observes all intermediates steps of HPO (and more); but somehow protects the privacy of the underlying model?

Also, the paper does not sufficiently explain its design choices. For example, on Page 6, the Knowledge Gradient come out of the blue; with no description of available methods and why this method was chosen. I also did not find the introduction of Bayesian Inference and in particular, the acquisition function to be standalone. I would have had a hard time following the paper had I not seen most concepts presented in this work before hand.

Finally I do not understand the placement of the related work in the middle of the paper (Section 5).

**Novelty.** I struggle to find the key novel contribution of the paper. Is it that the Pareto front is S-shaped? Is it that the paper uses this to reduce the number of queries made? Or is the main contribution the elicitation? I found the first two to be pretty incremental to Avent et al 2020. The last one would hold water if there was sufficient discussion of the "user model." I recognize that authors acknowledge this limitation, but for a paper built on pairwise elicitation this is rather unacceptable; especially empirically where the user model can skew results one way vs. another. This needs to be made clear front and center.

**Usability.** I am unclear who is the intended audience for this work. There is discussion of this in the paper where authors answer this question as the "privacy expert" and not the non-technical audience. I find this very puzzling. Why should the expert resort to inefficient pairwise comparisons to decide a utility-privacy trade-off? Especially if the same expert is running the HPOs? This paper should target the decision-maker who uses this tool as a decision-support system. I, however, understand that in its present form, the paper is not suitable for such an audience given the amount of background information the paper requires of them in Algorithm 1. This ties rather clearly with the lack of a "user model."

**Requested Changes:**

I have provided concrete recommendations above. To reinforce key ones:
- Please provide a concrete user-model. Justify it sufficiently and add experiments with other user models (preferably from the literature). Also some experiments with "worst-case" user models would be appreciated to check for robustness of the approach.
- Please add a hierarchical modeling graph to make the (co)-dependencies between $w$ and $\beta$ (or lack thereof) clear.
- Please argue for your threat model clearly in a separate section.
- Please consider re-writes to address presentational issues raised.

---

> ### Author Response · Authors · 2026-06-30
> **Response to reviewer 5xT9 -- part 1**
>
> Thanks for your review. We have uploaded a revision and highlighted changes to the original submission in blue.
>
> **Accuracy**
>
> > 1. Perhaps the most important one is the “fact that the privacy level in DP naturally serves as a direct optimization constraint”....Assuming that we can tightly optimize accuracy under a privacy constraint is therefore generally not correct.…I also believe this characterization is at odds with the empirical section where it is clear that many models are trained with various privacy bounds to achieve better accuracy.
>
> We thank the reviewer for pointing out this ambiguity. We believe the concern stems from two related but distinct issues.
>
> First, our work considers **theoretical differential privacy guarantees**, rather than empirical privacy measures such as membership inference attack success. We agree that the privacy parameter is a worst-case upper bound on privacy leakage and does not tightly characterize empirical privacy risk. Our intention was not to suggest otherwise. Rather, we treat the specified theoretical privacy budget as the optimization constraint, following the standard formulation in the differential privacy literature: for a fixed privacy guarantee, we optimize model accuracy over the remaining hyperparameters. To avoid this ambiguity, we revised the manuscript to replace phrases "*the privacy level serves as a direct optimization constraint*" with "*the theoretical privacy guarantee serves as the optimization constraint*" on Page 1, 2 and 6.
>
> Second, we believe there may be a misunderstanding regarding our experimental procedure. We do **not** optimize across different privacy budgets to obtain the highest accuracy. Instead, for **each fixed privacy level**, we independently perform HPO to identify the highest achievable accuracy under that particular privacy guarantee. The resulting privacy–accuracy pairs form the Pareto front, which is subsequently used in our preference-learning framework. We clarified this workflow in the revised manuscript on Page 11.
>
> > 2. Also, I think both $\beta$ and $w$ decide the Pareto front. The interaction between the two is unclear….
>
> We believe there may be a misunderstanding regarding the roles of $\beta$ and $w$. The Pareto front is parameterized solely by $\beta$, whereas $w$ parameterizes the decision-maker's utility function and determines the preferred point on the Pareto front. As defined in Section 2.2, the Pareto front depends only on the objectives, not on the decision-maker's preferences, while the most preferred trade-off jointly depends on both $\beta$ and $w$.
>
> > 3. Finally, I do not really follow the theoretical jump from a "decision release mechanism" in Page 7 to the "gradient release mechanism" of DP-SGD in the later sections of the paper.....
>
> We added the clarification on Page 8: **Our intention was not to claim that the theoretical derivation directly applies to DP-SGD.** Rather, as discussed in Section 4.1, we use the theoretical result as motivation for a broader structural hypothesis, which we validate empirically on DP-SGD-based models. As shown in Figures 3 and 5, both Gompertz and Logistic functions provide good approximations across the evaluated datasets.
>
> As highlighted on Page 14, we interpret the PatchCamelyon results differently. The S-shaped models still fit well, and they demonstrate that PACE does not require observing the full S-shaped curve. A local sigmoidal approximation over the application-relevant privacy range is sufficient to accurately model the trade-off, highlighting the flexibility of PACE.
>
> **Presentation**
>
> > 1. I found claims that contradict each other. Chief among them is the claim that HPO process does not consume privacy budget on Page 11 whereas on Page 4 authors cite Steinke and Papernot 2022 and other work that precisely model this privacy leakage…..
>
> We believe this concern arises from the different threat models considered. We clarified our threat model on Page 6 to justify this trusted curator choice and explained why the HPO process does not consume privacy budget in our setting.
>
> On Page 4, we cite Papernot & Steinke (2022) as background on private HPO, where the hyperparameter selection process or intermediate outputs must themselves satisfy privacy guarantees. In contrast, our experiments adopt the trusted-curator threat model of Avent et al. (2020), where model development is performed internally and only the final DP model is released. This trust-curator model is a standard setting in Dwork & Roth (2014). Under this threat model, the privacy guarantee protects the individuals in the training dataset against inference from the released model, not against the trusted internal curator. As discussed on Page 11, recent work (Xiang et al., 2024; Pradhan et al., 2025) has investigated the impact of non-private HPO on privacy and found no statistically significant evidence that it increases a modern deep learning model’s vulnerability to membership inference attacks.

---

> > ### Author Response · Authors · 2026-06-30
> > **Response to reviewer 5xT9 -- part 2**
> >
> > > 2. Also, the paper does not sufficiently explain its design choices. For example, on Page 6, the Knowledge Gradient come out of the blue;.... I also did not find the introduction of Bayesian Inference and in particular, the acquisition function to be standalone….
> >
> > As discussed on Page 6, we adopt the knowledge gradient because it is a one-step Bayes-optimal acquisition function. We compared it with other methods in the experiments section. We introduced Bayesian inference and acquisition function on Page 4.
> >
> > > 3. Finally I do not understand the placement of the related work in the middle of the paper (Section 5).
> >
> > We have moved it to Section 6, before the discussion section.
> >
> > >  **Novelty.** I struggle to find the key novel contribution of the paper. Is it that the Pareto front is S-shaped? Is it that the paper uses this to reduce the number of queries made? Or is the main contribution the elicitation?.....
> >
> >
> > We would like to kindly note that the acceptance criteria of TMLR are 1) Are the claims made in the submission supported by clear evidence? 2) Would some individuals in TMLR's audience be interested in the findings of this paper? Nor should work be rejected as “not novel enough,” since novelty is not a required criterion for acceptance. We have clarified our claims above, and the reviewer marked “Yes” for the second criterion.
> >
> > That said, our method is, to the best of our knowledge, the first to apply user preference modeling to the problem of choosing a preferred privacy–accuracy trade-off. Specifically, our contributions are (1) a theoretical characterization of the privacy–accuracy trade-off for logistic regression, which motivates modeling the Pareto front with S-shaped functions; (2) a Bayesian framework that explicitly models the Pareto front, enabling curve-based rather than pointwise preference elicitation; and (3) a more sample-efficient framework for both Pareto front learning and preference elicitation, thereby reducing both model evaluations and user queries. We clarified the contribution list in the revised manuscript on Page 3.
> >
> > Regarding the user model, we adopt the widely used Boltzmann-rational model rather than proposing a new preference model. Our contribution lies in the interaction framework, which is agnostic to the specific user model.
> >
> > > **Usability.** I am unclear who is the intended audience for this work….Why should the expert resort to inefficient pairwise comparisons to decide a utility-privacy trade-off?....
> >
> > Our intended user is a decision-maker (e.g., data practitioners and researchers) who is familiar with interpreting privacy budgets (e.g., $\varepsilon$) and is responsible for deciding how much privacy can be provided in a particular application while maintaining acceptable accuracy. We clarified this distinction in the revised manuscript on Page 16. As discussed in Section 7, explaining privacy budgets to non-expert users is an orthogonal problem that has been studied extensively in prior work.
> >
> > Regarding the interaction, we believe there might be a misunderstanding—our proposed method is **not** inefficient pairwise comparisons, but more efficient curve-based preference elicitation.
> >
> > **Requested changes:**
> > > 1. Please provide a concrete user-model. Justify it sufficiently and add experiments with other user models…..
> >
> > In our paper, the user model refers to the Boltzmann-rational choice model (Eq. 7) on Page 5, which is a standard choice in preference learning (Astudillo et al., 2023, Yamagata et al., 2024). Our contribution is a preference elicitation framework built on top of this standard choice model and extends it from discrete pairwise comparisons to continuous curve-based elicitation. Figures 8 and 9 illustrate the effect of different utility functions, while Figure 7 in the ablation study evaluates the proposed curve-based interaction against pointwise comparisons.
> >
> > > 2. Please add a hierarchical modeling graph to make the (co)-dependencies between $\beta$ and $w$ (or lack thereof) clear.
> >
> >
> > We would like to clarify that we assume the decision-maker's latent preference $w$ is independent of the Pareto front parameters $\beta$. This is a standard assumption in preference learning (Astudillo et al., 2023, Ozaki et al., 2024), where the user's preferences are treated as fixed but unknown and are inferred through interaction. Consequently, $\beta$ characterizes the Pareto front, while $w$ characterizes the decision-maker's preferences, which remain unchanged throughout the interaction. As shown in Figure 1, $w$ can affect which points on the front would be evaluated but the oracle front (red curve) remains independent of the preferences. Figure 9 further illustrates this by showing different preferences over the same trade-off curve. Modeling the relationship between $w$ and $\beta$ might be an interesting cognitive science problem, which is beyond the scope of our work.

---

> > > ### Author Response · Authors · 2026-06-30
> > > **Response to reviewer 5xT9 -- part 3**
> > >
> > > > 3. Please argue for your threat model clearly in a separate section.
> > >
> > > We added the clarification in the Section 3: problem statement on Page 6. The detailed response is provided above.
> > >
> > > > 4. Please consider re-writes to address presentational issues raised.”
> > >
> > > We have fixed the grammar issues in the revised version, and introduced PACE-L and PACE-G on Page 9.
> > >
> > > **References**
> > >
> > > Papernot & Steinke, 2022. Hyperparameter tuning with Renyi differential privacy. ICLR.
> > >
> > > Avent et al., 2020. Automatic discovery of privacy–utility Pareto fronts. PoPETS.
> > >
> > > Dwork & Roth, 2014. The algorithmic foundations of differential privacy. Foundations and trends in theoretical computer science.
> > >
> > > Xiang et al., 2024. Revisiting differentially private hyperparameter tuning. arXiv preprint arXiv:2402.13087.
> > >
> > > Pradhan et al., 2025. Hyperparameters in score-based membership inference attacks. SaTML.
> > >
> > > Astudillo et al., 2023. qEUBO: A decision-theoretic acquisition function for preferential Bayesian optimization. AISTATS.
> > >
> > > Yamagata et al., 2024. Relatively rational: Learning utilities and rationalities jointly from pairwise preferences. InICML 2024 Workshop on Models of Human Feedback for AI Alignment.
> > >
> > > Ozaki et al., 2024. Multi-objective Bayesian optimization with active preference learning. AAAI.

---

> > ### Comment · Reviewer_5xT9 · 2026-07-03
> >
> > > Second, we believe there may be a misunderstanding regarding our experimental procedure. We do not optimize across different privacy budgets to obtain the highest accuracy. Instead, for each fixed privacy level, we independently perform HPO to identify the highest achievable accuracy under that particular privacy guarantee. The resulting privacy–accuracy pairs form the Pareto front, which is subsequently used in our preference-learning framework. We clarified this workflow in the revised manuscript on Page 11.
> >
> > Well, this is less convincing. As you decrease the privacy budget, through HPO, you can likely find models with similar or sometimes better accuracy levels, which suggests that the previous privacy budget was too loose a bound. In other words, the new result can Pareto dominate the prior result. If you are not applying a Pareto operation after a privacy + HPO sweep, I do not think you are deriving the correct Pareto fronts.

---

> > > ### Author Response · Authors · 2026-07-03
> > > **Response to the follow-up question**
> > >
> > > We agree that the empirical HPO results are not necessarily monotonic and may contain dominated points due to stochastic optimization, finite HPO budgets, and training variability, as can also be observed in Figures 3 and 5. Our framework explicitly accounts for this. As discussed at the top of Page 9 in our paper,
> > > > "*In practice, the accuracy of any single model is a random variable due to stochastic elements in the training process and the noise inherent to the differential privacy mechanism. We adopt a probabilistic approach to account for this variability. In particular, we assume noisy observations are drawn from a Gaussian distribution whose mean is the expected accuracy predicted by the S-curve.*"
> > >
> > > Accordingly, our framework models the observed accuracies as noisy samples from an underlying latent privacy–expected accuracy trade-off (Eq. 18), rather than treating the empirical observations themselves as the Pareto front. **Therefore, Pareto filtering is unnecessary in our framework, since the observed accuracies are modeled as noisy samples from the latent trade-off rather than as Pareto-optimal points.** The S-shaped function serves as a probabilistic surrogate of this latent trade-off, allowing us to smooth over observation noise and efficiently identify the decision-maker's preferred operating point.

---

> > > > ### Comment · Reviewer_5xT9 · 2026-07-03
> > > > **accept with revisions**
> > > >
> > > > I thank the authors for their rebuttal and the interaction. I think most of the feedback has been addressed.